# Lava dome cycles reveal rise and fall of magma column at Popocatépetl volcano

Sébastien Valade [1] ✉, Diego Coppola [2], Robin Campion[1], Andreas Ley[3], Thomas Boulesteix [4], Noémie Taquet[4], Denis Legrand[1], Marco Laiolo [2], Thomas R. Walter [5] & Servando De la Cruz-Reyna [1]

Lava domes exhibit highly unpredictable and hazardous behavior, which is why imaging their morphological evolution to decipher the underlying governing mechanisms remains a major challenge. Using high-resolution satellite radar imagery enhanced with deep-learning, we image the repetitive dome construction-subsidence cycles at Popocatépetl volcano (Mexico) with very high temporal and spatial resolution. We show that these cycles resemble gas-driven rise and fall of the upper magma column, where buoyant bubble-rich magma is extruded from the conduit (in ~hours-days), and successively drained back (in ~days-months) as magma degasses and crystallizes. These cycles are superimposed on a progressive decadal crater deepening, accompanied by heat and gas flux decrease, which could be partially explained by gas depletion within the magma plumbing system. Results reinforce the idea that gas retention and escape from the magma column play a key role in the short- and long-term morphological evolution of low-viscosity lava domes and their associated hazards.

Lava domes essentially result from the extrusion of viscous lava outside a volcanic conduit, which accumulates near the erupting vent due to its high viscosity. Behind this general definition, however, hides a wide variety of dome morphologies, which are controlled by a combination of magma rheology, extrusion rate, and substrate topography[1]. Morphologies span from tall and steep domes (i.e., peléan domes with spine extrusions) to flat and circular domes (i.e., axisymmetric[2], pancake-shaped or "low" lava domes[3]), with a broad variety of intermediate shapes and extrusive features[1,2,4]. These two end-members are thought to be driven by two different growth mechanisms: exogenous (i.e., extrusive growth, whereby magma extrudes through the dome surface and generates discrete lava spines or lobes), and endogenous (i.e., intrusive growth, whereby magma causes internal swelling and expansion of the dome's outer carapace)[5]. The associated volcanic hazards, of both exogenously and endogenously growing domes, are highly dependent on the resulting dome

morphology[6], as instabilities may trigger rockfalls and collapses, potentially generating deadly pyroclastic density currents[7,8]. Moreover, rapid changes in the dome permeability can lead to sudden transitions from passive degassing to violent explosive events[9–13]. Due to the hazardous and unpredictable nature of growing lava domes, the study of dome morphology and extrusion dynamics have been essentially approached through modeling (analog[2,3,14] or numerical[15,16]) and remote sensing, including photogrammetry from ground-[17,18], air-[19,20], and satellite-based sensors using both optical and infrared bands[21–24]. However, lack of visibility is a major limitation to photogrammetry techniques. Synthetic Aperture Radar (SAR) is therefore a unique method to track dome growth and deformation, as radar microwaves penetrate clouds with little interference, therefore allowing imaging independently of cloud cover, volcanic steam or ash obstructing the view. Although interferometric processing (InSAR) has been widely used to quantify volcano-wide deformation[25] or even

[1]Universidad Nacional Autónoma de México, Instituto de Geofísica, Mexico City, Mexico. [2]Università degli Studi di Torino, Dipartimento di Scienze della Terra, Torino, Italy. [3]Department of Computer Vision & Remote Sensing, Technische Universität Berlin, Berlin, Germany. [4]Volcanology Research Group, Department of Life and Earth Sciences, Instituto de Productos Naturales y Agrobiología (IPNA-CSIC), La Laguna, Spain. [5]GFZ German Research Centre for Geosciences, Telegrafenberg, 14473 Potsdam, Germany. ✉e-mail: valade@igeofisica.unam.mx

summit deformation[26,27], the small spatial extent of domes and their tendency to be incoherent in InSAR imagery (due to rapidly changing surface and morphology) have made this technique less suited for studying lava dome emplacement. Instead, the reflected SAR intensity has been used to track the dome and crater morphology[21,28–31]. Nonetheless, while SAR imaging has the advantage of sensing through clouds, visual interpretation of the resulting intensity images is hindered by both the intrinsic radar viewing geometry and intrinsic granular noise (speckle). Indeed, unlike geocoded images which are displayed with respect to cardinal directions, images in radar geometry are displayed with respect to the sensor itself, i.e., along the satellite motion direction ("azimuth", image y-axis), and along the radar look direction ("range", image x-axis). Although such imaging geometry is subject to geometric distortions (i.e., slopes facing towards the radar will appear bright and compressed, and slopes facing away will appear dark and stretched), it can be used to our advantage to recover quantitative measurements of the surface topography. In particular, slopes facing away from the satellite with an angle steeper than the radar incidence angle will cast a shadow, from which vertical depths can be estimated from trigonometry[29,31]. The method is applied here to track the variations of Popocatépetl volcano (Mexico) inner-crater depth (Fig. 1, "Methods"), which is particularly difficult to access due to its >5400 m high summit and highly hazardous explosive activity.

Popocatépetl has experienced since its reactivation in 1994[32] successive episodes of lava dome construction and destruction[33,34]. The morphological characteristics of the domes fall into the broad category of low lava domes[3], typically pancake-shaped and affected by subsidence and dome-destruction explosive processes following their emplacement[34]. Prior to the eruption onset, the crater floor hosted a small lake which evaporated before the initial phreatomagmatic activity in December 1994. At that time, the lowest level of the crater was about 4940 m above sea level. The lava extrusion activity began in March 1996 with a series of dacitic lava dome emplacements and destructions that slowly filled with lava fragments and pyroclastic debris a large part of the main crater[35]. After the emplacement of the largest domes in 2000–2003, the rate of destruction slowly exceeded the rate of debris accumulation. This resulted in the formation of an inner crater, surrounded by a terrace that almost reaches the lowermost sector of the main crater rim, at an altitude of approximately 5120 m above sea level[34]. Subsequently, several domes grew within this inner crater, and this study focuses on the domes emplaced between 2012 and 2020. For simplicity, we call any lava extrusion above the 1994 crater floor a "dome", even if it is confined within the inner crater. The repetitive dome growth episodes are thought to result from varying buoyancy of the magma column, induced by varying dissolved volatile proportion in the magma[36]. Popocatépetl is also characterized

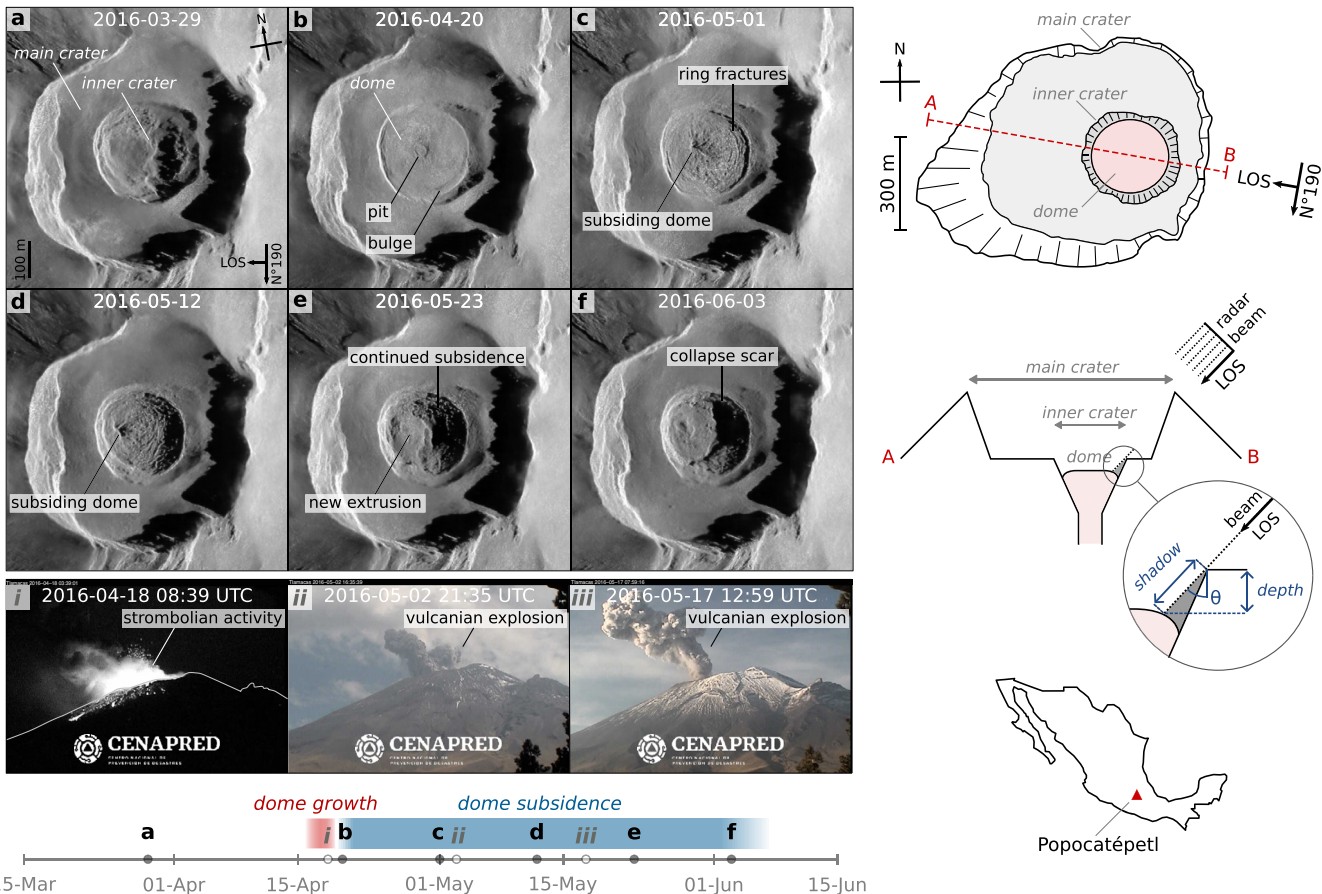

**Fig. 1 | Typical dome construction and destruction cycle viewed from satellite TerraSAR-X (TSX) imagery and surveillance cameras.** TSX intensity images are displayed in radar geometry: horizontal axis corresponds to radar line-of-sight LOS direction (radar looking from right to left), vertical axis corresponds to satellite heading direction (N°190). **a** TSX acquisition prior to the dome emplacement; **b** acquisition shortly after the dome emplacement: the dome starts to deflate and drain back into the conduit, as evidenced by the peripheral bulge and central pit; **c, d** the dome is subsiding, as evidenced by the development of ring fractures and deepening of the crater (increasing shadow width and migration of the central depression away from the radar, i.e., towards the left of the image); **e, f** the crater floor continues to deepen through stepwise piston-collapse structures accommodated by concentric fractures, leaving semicircular collapse scars. CENAPRED surveillance camera snapshots of: **i** strombolian activity during dome growth, and **ii–iii** vulcanian explosions during dome subsidence. Top-right sketch crater contours are based on a high-resolution satellite image acquired on 2016-05-07 (i.e., 6 days after the TSX acquisition shown in (**c**)) available from Google Earth. Middle-right sketch shows how radar shadow is used to recover crater depth from trigonometry using the radar incidence angle θ (sketch not to scale).

by strong excess degassing[37,38], as the emitted mass of $SO_2$ largely exceed the mass dissolved in the erupted magma. This characteristic has led authors to suggest that unerupted magma is degassing either from a deeper magma reservoir through a gas-permeable conduit[39], or from a convecting magma column at shallower depths[40].

Here we investigate the shallow magma dynamics governing the repetitive dome cycles and peculiar excess degassing at Popocatépetl using multiparametric satellite datasets. We use high-resolution TerraSAR-X (TSX) and medium-resolution Sentinel-1 (S1) SAR images acquired over 8 years (2012–2020) to: (1) quantify vertical variations of the inner-crater depth, revealing both short-term dome construction-subsidence cycles and long-term crater deepening and widening, and (2) analyze dome morphological evolution, revealing magma emplacement and withdrawal mechanisms with exceptional temporal and spatial detail, thanks to a deep-learning image enhancement approach. We compare this data with 15 years (2005–2020) of $SO_2$ gas emission and infrared thermal radiation observations from the satellite sensors OMI (Ozone Monitoring Instrument) and MODIS (Moderate Resolution Imaging Spectroradiometer), which confirm that magma volumes required to sustain the gas and thermal fluxes largely exceed the actual erupted magma volumes. This unique combination of observations offers a new comprehensive view of the eruptive dynamics and magma conduit processes operating at Popocatépetl, where gas retention and escape could explain the short- and long-term ups and downs of the magma column. The results are consistent with observations at other low-viscosity lava dome volcanoes, and point to unexpected similarities with magmatic processes operating in the upper magma column of more basaltic open-systems. This study opens new perspectives to constrain the overarching characteristics of open-vent volcanic activity[41,42], and paves the way to improved multidisciplinary satellite volcano monitoring[43] and hazard assessment.

## Results

### Dome construction-destruction cycles

The analyzed satellite SAR imagery provides a dense temporal and spatial view of the summit crater since the radar sensor is insensitive to obstructing clouds and volcanic plumes. The granular noise intrinsic to raw intensity images is here filtered with a specifically designed convolutional neural network, thereby revealing morphological details otherwise hardly visible in unfiltered images (see "Methods" and Supplementary Figs. 1–3). Numerous cycles of dome construction and destruction are identified, outlining a pattern with repeating morphological features. Throughout the 8-year dataset (2012–2020), domes were successively emplaced and destroyed within an inner-crater of the main crater (Fig. 1, Supplementary Video 1). We hereafter use the April 2016 dome as a case example, and refer to Supplementary Figs. 5 and 6 for additional showcases. The dome construction phase appears as a fast process (i.e., lasting a few hours to days), usually captured by ≤1–2 TSX acquisitions only. The dome morphology is typically pancake-shaped (Fig. 1b), i.e., sub-circular with a low height/diameter ratio, and a nearly flat and smooth profile. The measured diameter ranges between ~45–270 m, and the measured thickness at the edges between ~3–6 m. In some occasions lobes are observed near the dome center, suggesting pulses during emplacement. In rare cases, annular extensional fractures can be distinguished. Considering the geometric assumptions of the inner-crater walls (see "Methods"), extruded dome volumes range between ~0.05–2 Mm³ (±0.5 Mm³). Ground-based observations indicate that dome growth episodes are commonly accompanied with strombolian-like activity, characterized by continuous emission of ash and incandescent ballistics[11] (Fig. 1i).

Dome destruction on the other hand is controlled by two distinct mechanisms. The first, is the progressive subsidence of the dome surface, during which the dome deflates and sinks back into the volcanic conduit. The subsidence starts shortly after the dome emplacement (within the first 2 days according to Fig. 1i and b), and lasts several days-months. In the early stages, a circular pit in the center of the dome is occasionally observed, with diameters ranging between ~23–47 m (Fig. 1b). At the same time, bulges can be seen at the dome periphery, reflecting the collapse of the inner parts of the dome. A recurrent feature is the development of ring fractures on the dome surface which accommodate the downward sag (Fig. 1c, d). As the dome progressively deepens, the central depression moves away from the radar sensor (i.e., towards the left in TSX images, Fig. 1c, d). In the late stages of the subsidence, piston-collapse structures are sometimes visible, with characteristic semicircular faults and flat floor (Fig. 1f). In addition, because the inner-crater walls are left unstable, they are subject to landslides which occasionally leave visible detachment scarps that contribute to the incremental inner-crater enlargement (Supplementary Fig. 6).

The second mechanism contributing to lava dome destruction are the vulcanian explosions commonly recorded in the days-weeks following the emplacement[11,34] (Fig. 1ii–iii). The impact of these explosions on the dome morphology is, however, more difficult to estimate from SAR images, as several can occur between consecutive TSX acquisitions. Large ballistic blocks landing on the main crater floor are sometimes visible, and most likely result from the dome fragmentation. After powerful explosions, a depression is clearly identifiable, though with more irregular contours and depths than the piston-collapse structures described above (Supplementary Fig. 6).

### Crater deepening and widening

Alongside the repeated cycles of dome construction and destruction, a progressive enlargement and deepening of the inner-crater is observed throughout the past decade (Fig. 2a–c), which contrasts with the progressive infilling observed in the years following the 1994 eruption onset[35]. Early 2012, domes were emplaced within a crater about ~220 m wide and a few meters deep, such that domes were nearly at the same level as the main crater terrace. By late 2019, a crater ~350 m wide and ~150 m deep had developed. This deepening and widening of the inner-crater is accompanied by a progressive decrease in thermal Volcanic Radiative Power (VRP) recovered from the MODIS satellite images (by ~20 MW, Fig. 2d), as well as a decrease in the monthly $SO_2$ flux recovered from OMI satellite images (from $12.8 \times 10^3$ tons/day in May 2012 to ~500 tons/day in 2020, Fig. 2e). The volume loss caused by the crater excavation during this 8-year period represents ~8 Mm³ ± 2 Mm³ (Fig. 3b), assuming that the inner-crater shape is a truncated cone with average slopes of 60° ± 20° (Fig. 3a). Inner-crater volume gains (i.e., extrusion volumes $\Delta V_c+$) and losses (i.e., excavation volumes $\Delta V_c-$) are estimated by taking, respectively, positive and negative differential crater volumes $\Delta V_c$, and show that from mid 2016 onwards the excavation rate increases with respect to the extrusion rate (Fig. 3c). The main crater floor on the other hand was progressively filled by pyroclastic deposits, with a vertical rise of ~15 m (±5 m) between early 2012 and late 2019. Considering the main crater terrace area late 2019 (~0.19 km² estimated from a high-resolution satellite image acquired on 12/07/2019 available on Google Earth), this results in a very rough estimate of 2.8 Mm³ (±1 Mm³) of pyroclasts deposited on the main crater floor during ~8 years, which is far less than the inner-crater volume loss.

### Heat, gas, and extrusion correlations

Volcanic heat radiation VRP shows no clear correlation with lava extrusion volumes (i.e., dome volumes, Supplementary Fig. 7). Instead, sporadic high VRP values are associated with explosive events, which destroy the dome, expose the hot core to the atmosphere, and deposit incandescent pyroclasts on the volcano flanks[23]. In the long-term, however, thermal radiation shows a striking correlation with gas fluxes. Figure 4b shows the coincident variations of VRP and $SO_2$ fluxes recorded over a 15-year period (2005–2020), smoothed with a 365-day running average to remove both seasonal and eruptive transients. The

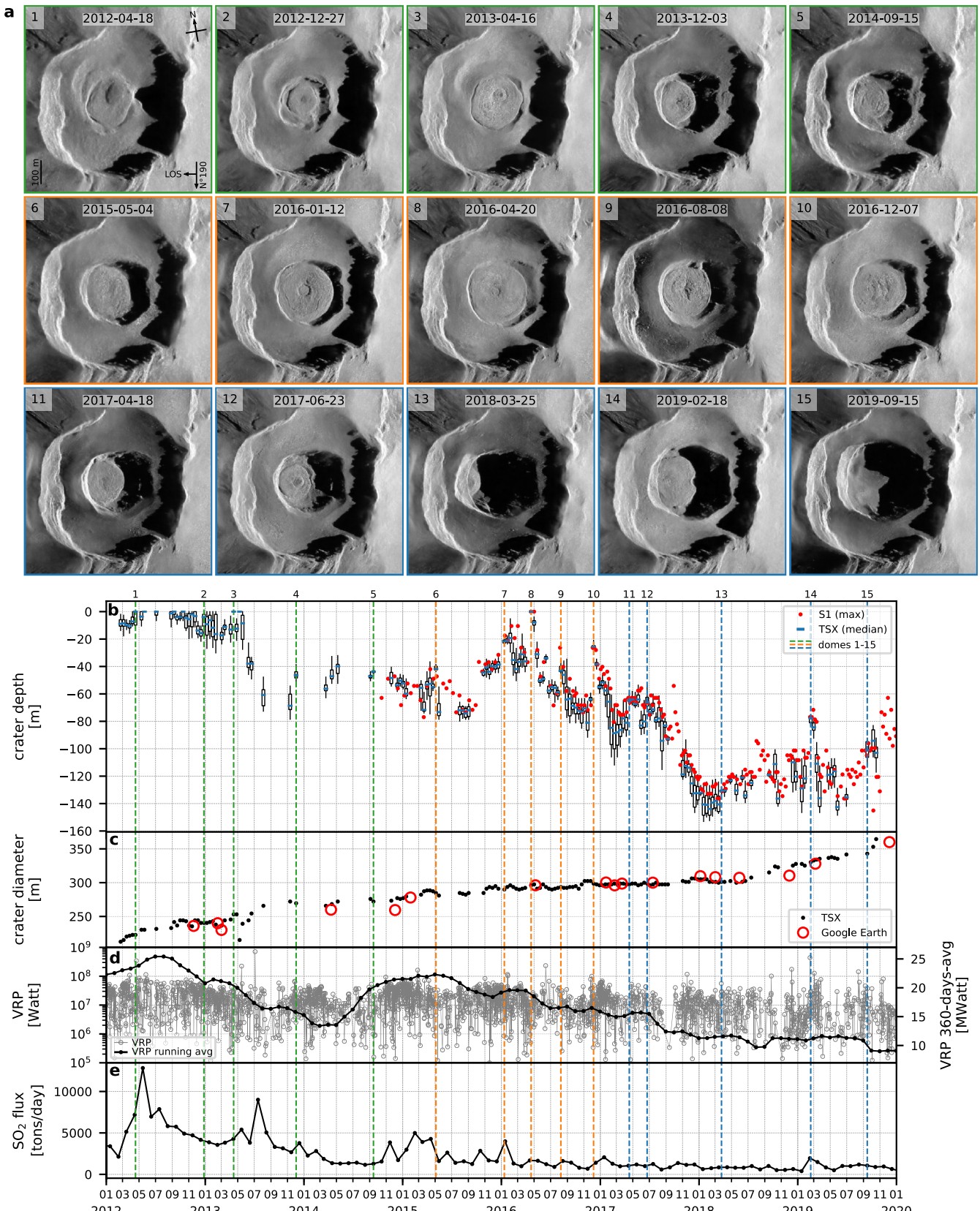

**Fig. 2 | Crater deepening and widening alongside progressive decrease in heat and gas fluxes. a** TerraSAR-X (TSX) images showing a selection of 15 lava domes. Dome growths are associated with a reduction of the inner-crater depth. **b** Inner-crater crater depth variations estimated from TSX and Sentinel-1 (S1) images (see "Methods" for description of the box-plot elements). **c** Inner-crater crater diameter estimated from TSX and Google Earth images. **d** Thermal volcanic radiative power VRP estimated from MODIS images. **e** Monthly $SO_2$ flux estimated from OMI images.

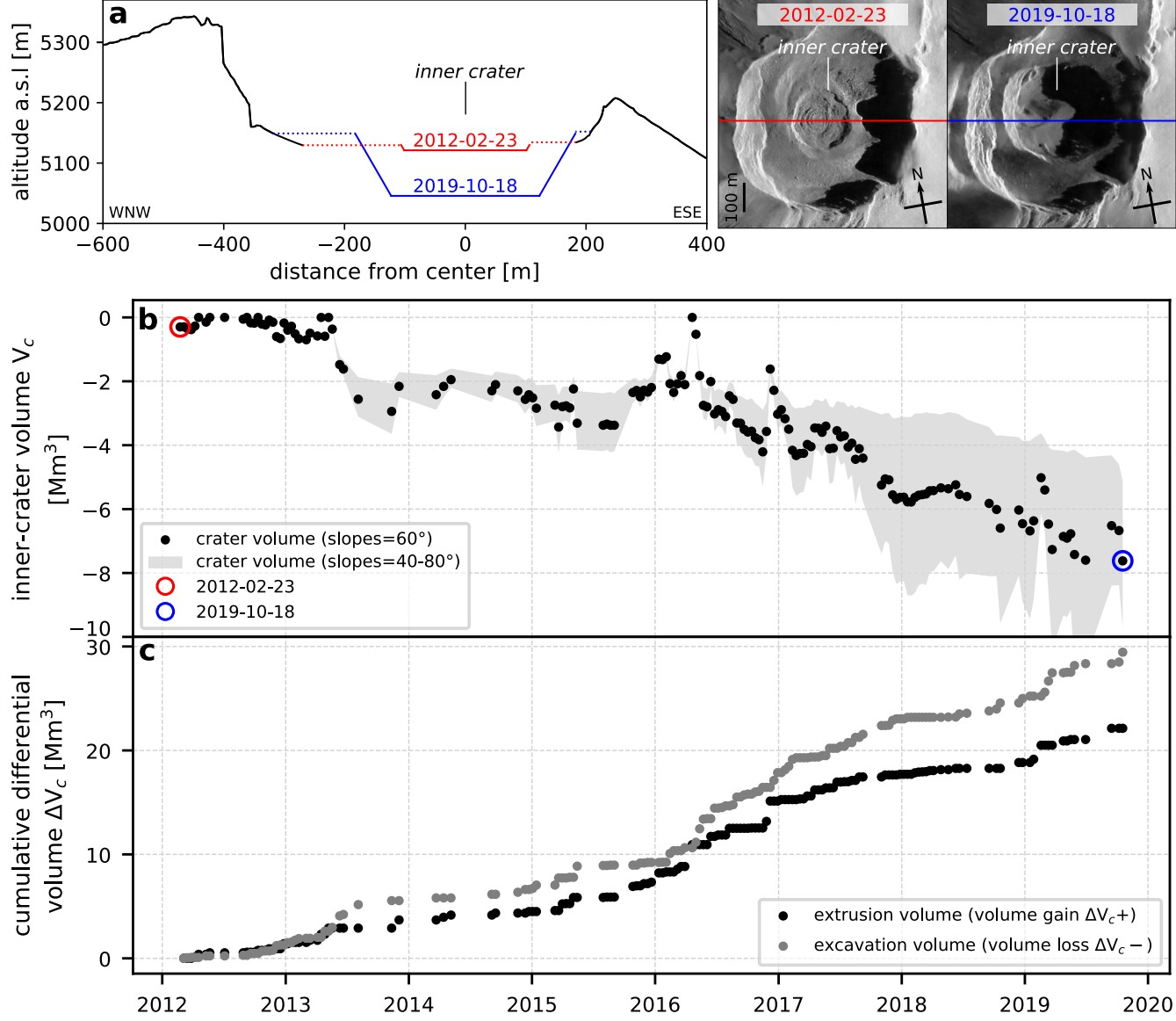

**Fig. 3 | Inner-crater volume loss. a** Cross-sections showing the deepening and widening of the inner-crater between early 2012 and late 2019: crater diameter and depth are calculated from TSX images, and crater shape is displayed as a truncated cone with 60° slopes. **b** Inner-crater volume $V_c$ variations, calculated for a range of assumed crater slopes (60° ± 20°). **c** Cumulative inner-crater extrusion volumes (crater volume gain $\Delta V_c > 0$) and excavation volumes (crater volume loss $\Delta V_c < 0$).

cross-plot of VRP and $SO_2$ fluxes (Fig. 4c) shows that the thermal radiation is linearly correlated with the gas flux, with a best-fit ($R^2 = 0.85$) given by: VRP = $10.5 \cdot SO_2 + 1.8$, where VRP is expressed in MW and $SO_2$ flux in kilotons/day. The period associated with high $SO_2$/VRP ratios (i.e., $SO_2$/VRP > 0.1 corresponding to Aug-2011 to Nov-2015, Fig. 4a) was ignored in the calculation of the best-fit, as it is decoupled from the rest of the trend due to unusually high gas emissions with respect to the thermal radiation. This period is associated with what is believed to be a new injection of juvenile magma into the system[44], which is reflected here by an increase in gas flux much stronger than the increase in thermal radiation.

**Magma budgets**

Magma volumes coming in and out of the system are estimated from the satellite dataset to infer on the subsurface magma dynamics. The magma input supply rate in particular is inferred from the $SO_2$ gas emissions, as the detected mass of gas is related to a given volume of magma degassing at depth[37]. The magma output on the other hand is harder to quantify, as it depends on both the volume of extruded lava

which radiates thermally (and part of which is drained back in the conduit), and the volume of lava expelled outside the crater as tephra products (ash and ballistics). We hereafter quantify the shallow magma budget following ref. 21 (see "Methods" for details), by estimating the following magma volumes (Fig. 5a): volume of degassed magma ($V_{degas}$) derived from the $SO_2$ emissions, volume of radiating magma ($V_{thermal}$) derived from the thermal VRP, and volume of extruded magma ($V_{extruded}$) derived from the SAR images. Because we have no estimate of the long-term volume of magma ejected as tephra outside the crater ($V_{tephra}$), we use a semi-empirical approach assuming that $V_{tephra} = 1/3 \cdot V_{extruded}$ (the factor 1/3 is approximated as the ratio of the total ejected material to the cumulative lava extruded, see ref. 34 and "Methods"). These volumes are then smoothed and converted to monthly averaged fluxes $Q_{degas}$, $Q_{thermal}$, $Q_{extruded}$, and $Q_{tephra}$ (Fig. 5b, "Methods"). The imbalance $V_{degas} \gg V_{extruded} + V_{tephra}$ by a factor of ~45 characterizes the "excess degassing"[37–39] observed at many open-system volcanoes, which indicates that there is more magma degassing than just the degassed erupted magma. On the other hand, the imbalance $V_{thermal} \gg V_{extruded} + V_{tephra}$ by a factor of ~15 characterizes

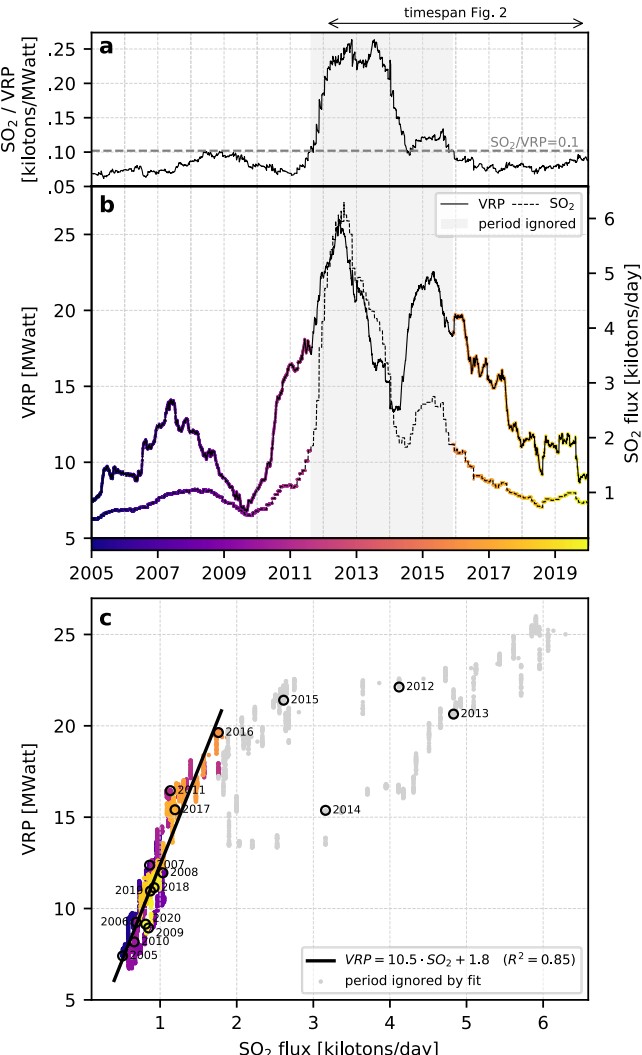

**Fig. 4 | Long-term correlation between thermal volcanic radiative power VRP and SO$_2$ gas flux. a** Ratio between SO$_2$ and VRP. **b** Time series of VRP and SO$_2$ flux. **c** Cross-plot between VRP and SO$_2$ flux, and best linear fit. The time period when the ratio SO$_2$/VRP > 0.1 (Aug-2011 to Nov-2015) is ignored in the calculation of the fit.

the "excess thermal radiation"[21], which indicates that there is more magma releasing heat than what is actually being erupted. These observations have important implications for the underlying magma system, as they show that the average magma input rate $Q_{in}$ (=$Q_{degas}$ = 2.55 m$^3$/s) required to sustain the measured gas emissions is far greater than the average magma output rate $Q_{out}$ (=$Q_{extruded}$ + $Q_{tephra}$ = 0.06 m$^3$/s). Likewise, the average magma flux required to sustain the measured thermal radiative power $Q_{thermal}$ is far greater than the average magma output rate $Q_{out}$. This suggests that unerupted magma is degassing and cooling at shallow levels in the plumbing system.

## Discussion

High-resolution satellite imagery combined with deep-learning enhancement techniques reveal unprecedented levels of detail in the morphology and structural evolution of low-viscosity lava dome cycles at Popocatépetl. The dome morphological characteristics suggest that the growth-subsidence cycles are similar to gas-driven rise and fall of the upper magma column. The construction phase is characterized by the extrusion of a low-viscosity magma, resulting in a pancake-shaped dome (Fig. 6a, b). The extrusion is most likely explained by the arrival of a gas-rich magma batch at the surface, causing (i) a rapid rise due to

increased buoyancy, (ii) elevated SO$_2$ and halogen emissions[11,45], (iii) surface strombolian activity[11], and (iv) fluid flow inside the conduit revealed by characteristic seismic tremor signal[34,46,47]. The destruction phase on the other hand is characterized by two processes: a progressive dome subsidence lasting weeks to months, which is punctuated by sporadic vulcanian explosions (Fig. 6c, d). The subsidence is likely related to the combined action of (i) the cooling and gas release from the emplaced magma body (foam collapse[12] and gravitational compaction[11], i.e., "soufflé" effect), and (ii) the drainage of extruded magma back into the conduit[12,24,48]. Evidence of this is provided by the subsiding dome center (occasionally materialized as a small pit which could reflect the underlying feeding conduit), the peripheral bulge, the development of concentric ring fractures accommodating the downward sag, as well as sub-circular piston collapse structures. Such ring fractures and piston-like subsidence of the crater floor have been previously described at Popocatépetl[34] and other volcanoes such as Láscar[12] (Chile) and Mount Cleveland[24] (USA). These combined observations remind a gas-driven upward and downward advection of magma: an initial increase in the gas content leads to an increased buoyancy resulting in magma extrusion from the conduit, and the subsequent gas/heat loss and crystallization leads to a buoyancy decrease resulting in the drain-back into the conduit. The subsidence, in turn, is thought to reduce the permeability of the system through temporary closure of gas escape pathways, promoting episodes of pressurization leading to vulcanian explosions[11,12]. Modeling of the magma column density[36] over a range of pressures, temperatures, and dissolved water contents, has shown that the column height is very sensitive to small changes in the dissolved volatile content, and it has therefore been suggested that the dome growth (and collapse) at Popocatépetl could be buoyancy-driven[34,36,49]. The dataset presented here provides further evidence supporting this idea, which we extend by suggesting that variations in the exsolved gas fraction of the upper magma column can explain the observed depth variations. Indeed, the dome emplacement is followed by rapid dome subsidence, reaching ~30–60 m drop in the ~30 days after dome emplacement (Fig. 6e). Although the vulcanian explosions which usually follow dome emplacement contribute to the crater excavation, we suggest that the compaction of the uppermost magma column due to outgassing of a foamy layer plays a key role in the observed crater subsidence. Indeed, the uppermost part of the magma column is expected to have very high exsolved gas bubbles fraction, possibly reaching 40–70 vol.%, according to vesicle size distributions analysis of juvenile clasts at Popocatépetl[50], numerical models of conduit dynamics in dome building eruptions[51,52], observations from muon-tomography of rhyolitic magma conduit[53], as well as petrologic and experimental evidence from basaltic[54] and rhyolitic[55] compositions, respectively. We suggest that the volume loss due to the release of this gas in a short period of time can explain the compaction of the magma column and the observed crater deepening. If we consider a cylindrical portion H$_f$ of the upper magma column filled with a bubble-rich magma, the magma level variation $\Delta H$ associated to changes in the volume gas fraction $X_{gas}$ can be simply recovered from $\Delta H = H_f(1-X_{gas0})/(1-X_{gas}) - H_f$, where $X_{gas0}$ is the initial gas fraction at the time of the dome emplacement, and R the conduit radius (see "Methods"). Following previous work[36] at Popocatépetl, we assume $X_{gas0} = 0.5$ and $H_f = 100$ m ± 75 m, and we consider for simplicity that the gas fraction decreases linearly until reaching values comparable to the mean porosity of erupted pyroclasts during dome-forming eruptions (i.e., 15–20%)[56,57]. Figure 6e shows that under such conditions, the magma level variations $\Delta H$ calculated during the first ~30 days can explain the observed dome subsidence rates (see corresponding TSX images in Supplementary Fig. 5). The sustained degassing fluxes with high HCl/SO$_2$ ratios observed after dome extrusions are in agreement with the degassing of a superficial magma[44,58]. Although this approach is simplistic as it ignores the effects of both the gravitational load due to the crystallized

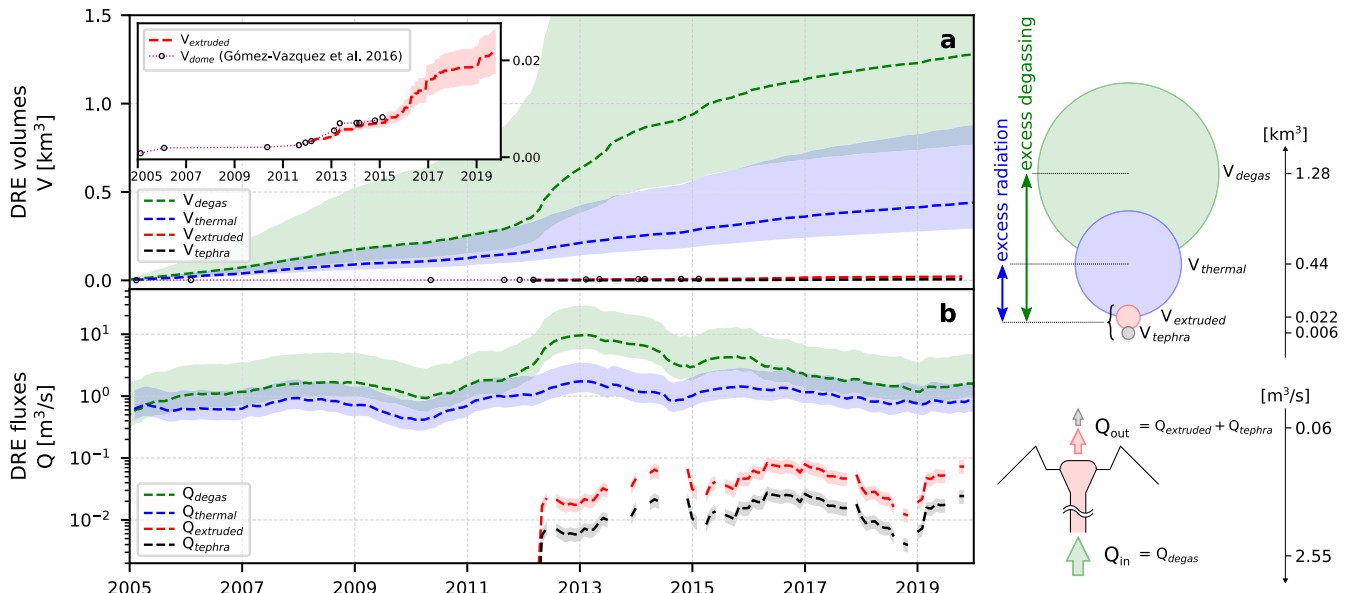

**Fig. 5 | Magma budget estimation. a** Cumulative dense rock equivalent (DRE) volumes of degassed magma $V_{degas}$, thermally radiant magma $V_{thermal}$, extruded magma $V_{extruded}$, and tephra $V_{tephra}$. **b** Magma fluxes $Q_{degas}$, $Q_{thermal}$, $Q_{extruded}$, and $Q_{tephra}$ derived from the corresponding volumes. $Q_{in}$ and $Q_{out}$ respectively represent magma fluxes coming in and out of the system. Colored envelopes represent the range of values calculated when considering extreme-case parameters (see "Methods"). Dome volumes prior to 2012 (i.e., prior to the TSX dataset) are taken from ref. 34.

portion of the dome, and the potential drainage of degassed magma back into the conduit, it shows that gas depletion and compaction of the upper magma column likely plays a major role in the crater deepening following the dome emplacements, as previously suggested at Láscar volcano[12].

If the above model seems able to explain the short-term variations in the crater depth associated with individual dome cycles, the progressive deepening and enlargement of Popocatépetl's inner-crater observed over the last decade raises the question of the driving mechanism, and its relationship with the dome construction-destruction cycles. Two processes can be considered: (i) the excavation due to repeated explosions, and/or (ii) the depressurization of the magma plumbing system due to persistent passive degassing and decreasing magma supply rates. Although both mechanisms likely operate and contribute to the progressive crater deepening, our observations suggest that the latter plays an important and previously unsuspected role (Fig. 7). Indeed, analysis of the daily reports compiled by the local volcano monitoring institution (CENAPRED) indicates that the long-term crater deepening rate does not show any clear and systematic correlation with the explosion rate nor with the emitted ash altitudes (Supplementary Figs. 8 and 9). Moreover, the infilling rate of the main crater by pyroclastic deposits (resulting from the repetitive explosive activity), does not mirror the inner-crater deepening rate (Supplementary Fig. 8a). However, the inner-crater deepening and widening is accompanied by a long-term decrease in both degassing rates and thermal radiative power, by 2 and 1 orders of magnitude, respectively, in 8 years. Intuitively, one might expect that the long-term, persistent degassing would induce a progressive densification of the magma column, which would be accommodated by the gravitational deepening of the crater as the magma column sank (Fig. 7a–c). To test the effect of continuous degassing on the depressurization of the magma reservoir and column and its possible role in the observed crater deepening, we apply the model of ref. 59. This model considers an idealized system where a magma reservoir is connected to an open cylindrical conduit filled with magma, where the reservoir pressure is magmastatic, and where the conduit pressure is subject to the same pressure changes as the reservoir. The model provides an analytical solution for the pressure change with time $\Delta P(t)$

in the column/reservoir, as a function of constant gas flux, constant conduit radius, initial reservoir volume, and magma/host rock properties. In turn, the magma level variations $\Delta H$ in the conduit expected from the pressure changes can be recovered from $\Delta P(t)/(g\rho_{m,c})$, where $\rho_{m,c}$ is the mean density of melt (i.e., bubble free magma) in the column, assuming that the mass of melt in the conduit is much larger than the mass of gas, and that the conduit radius and melt density are constant. We contemplate the model scenario in which gas exsolution occurs at low pressures via magma convection in the conduit[60], in agreement with both petrological studies at Popocatépetl[40] and excess degassing/excess thermal radiation reported here. The degassing rate is in turn directly controlled by this convection (parametrized in the model following refs. 38, 60), whereby the gas-rich magma upflow rate ($Q$) loses a fraction ($n_c$) of its dissolved volatile content, before sinking back into the conduit due to increased density. We here consider two end-member cases of the magma input flux $Q = 10$ m³/s and $Q = 1$ m³/s (i.e., $Q_{degas}$ inferred from the $SO_2$ fluxes calculated in 2013 and 2020, respectively) as input to the model. The conduit radius ($R$) is constrained from these magma input fluxes according to the convection parametrization[60], and result in $R = 14.8$ m and $R = 8.3$ m, respectively. The remaining parameters necessary to calculate $\Delta P(t)$ are fixed based on previous studies, namely the degassing scenario described by ref. 40 (see "Methods" and Supplementary Table 1 for details). The calculated pressure variations are then used to recover magma level variations $\Delta H$ in the conduit (Fig. 7d, dashed lines), which are compared to the observed crater deepening in the past 8 years (Fig. 7d, black markers). The observed crater depth variations are in the range of those predicted by the model in the two extreme cases of degassing rates, and on average closely follow the parametrization suggested by ref. 40 at Popocatépetl ($Q = 7$ m³/s). The overall agreement of this model with our observations therefore suggests that progressive depressurization due to persistent degassing could contribute to the progressive deepening of Popocatépetl's inner-crater observed over the past decade. The coeval crater widening which is observed, could instead be a direct consequence of this deepening, due in particular to the instability of the crater walls (frequently affected by gravitational landslides), as well as the repeated vulcanian explosions. Although the above mechanism is still speculative, the model suggests that within

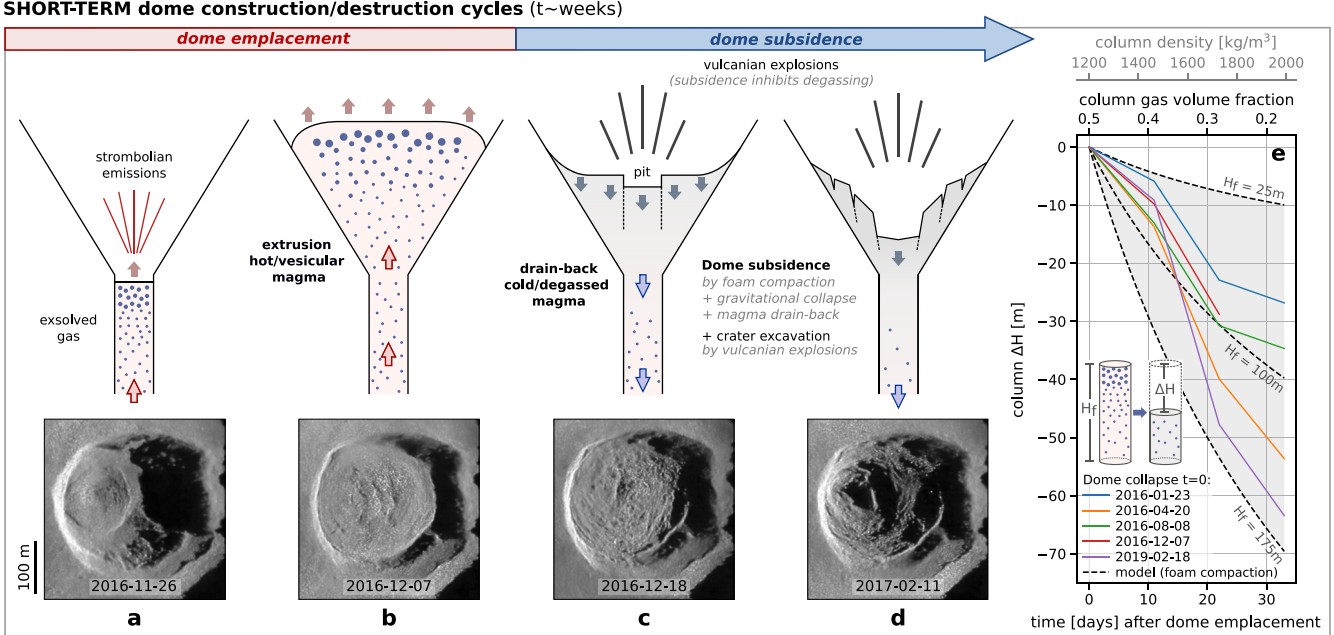

**Fig. 6 | Short-term dome construction-destruction cycle interpreted as the gas-driven rise and fall of the upper magma column. a, b** Extrusion of a hot-vesicular magma resulting in the lava dome emplacement, followed by **c, d** dome surface subsidence due to gas depletion, cooling, and magma drain-back in the conduit (sketches not to scale). Vulcanian explosions and gravitational collapse of the inner walls contribute to the crater deepening and enlargement in the days-weeks following the dome emplacement. **e** Dome subsidence modeled as resulting from the gas depletion (foam compaction/escape) of the upper portion $H_f$ of the magma column, and leading to a decrease in column height ΔH. Dashed black lines: model assuming $H_f = 100\,m \pm 75\,m$, and an initial gas volume concentration $X_{gas0} = 0.5$ decreasing linearly to reach $X_{gas} = 0.2$ in 30 days. Solid colored lines: selected dome subsidence trends, i.e., depths measured by TerraSAR-X (TSX) after the dome emplacement (t = 0), and normalized so that H = 0 corresponds to the hightest point reached by the dome (see corresponding TSX images in Supplementary Fig. 5).

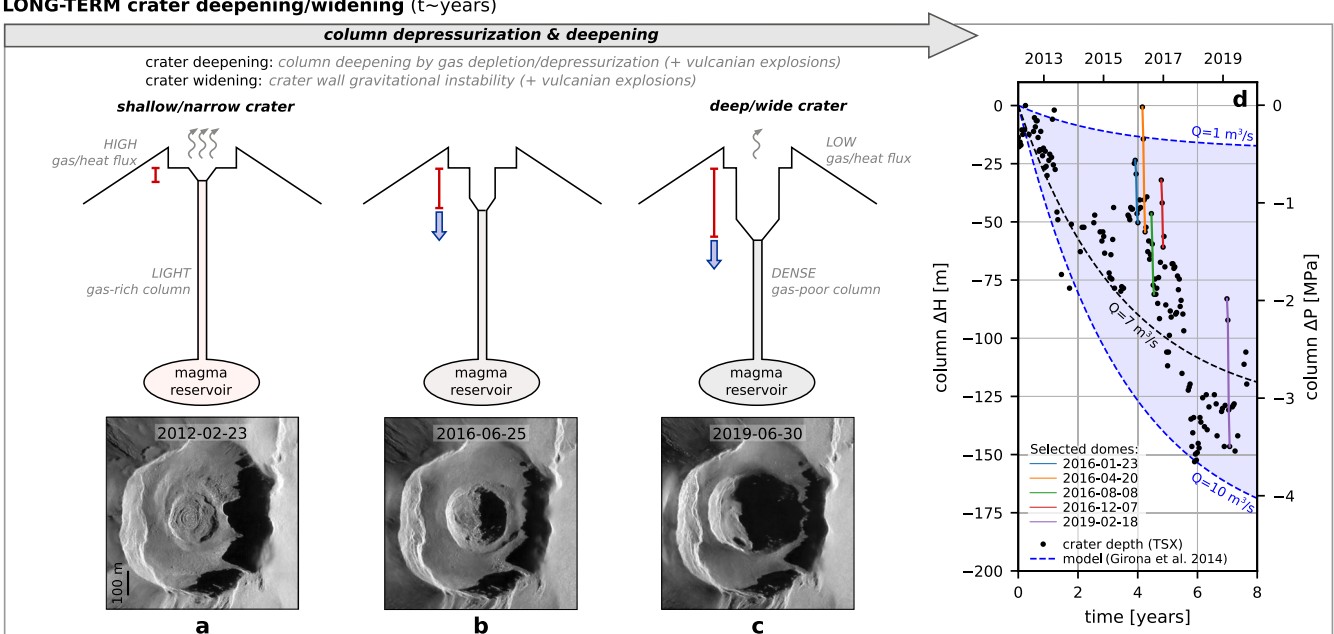

**Fig. 7 | Long-term crater deepening and widening (~8 years). a–c** Progressive crater deepening and widening, accompanied by continuous decrease in gas and heat fluxes. Crater deepening is depicted as driven by magma column drop due to progressive gas depletion and densification, while crater widening is thought to accompany this deepening by gravitational collapse of the inner wall (sketches not to scale). Vulcanian explosions likely contribute to both deepening and enlargement. **d** Model (ref. 59) of magma column and reservoir depressurization ΔP due to passive degassing, which results in variations of the magma column height ΔH. Dashed lines: model for various volumetric input magma fluxes Q (the other fixed input parameters are provided in Supplementary Table 1). Black markers: crater depths measured from TerraSAR-X (TSX) acquisitions (2012–2020). Colored lines: selected dome subsidence trends (see Fig. 6e and Supplementary Fig. 5).

reasonable input values and assumptions, it can predict the overall long-term crater deepening trend.

The novel combination of observations presented here, including high-resolution radar imagery of the crater's morphological evolution collected over 8 years (2012–2020), together with infrared thermal radiation and $SO_2$ degassing collected over 15 years (2005–2020), offers a new comprehensive view of the eruptive dynamics and magma conduit processes operating at Popocatépetl. The combined observations suggest that: (1) the short-term dome construction-subsidence cycles resemble gas-driven ups and downs of the uppermost magma column, where retention and escape of exsolved gas are responsible for magma extrusion and withdrawal from the conduit, (2) the long-term progressive inner-crater deepening and the concomitant decrease in heat/gas fluxes could reflect the progressive depressurization of the magma conduit and reservoir due to progressive gas depletion, (3) the excess degassing and excess thermal radiation of the system suggests that unerupted magma is degassing and cooling at shallow levels below the crater floor. These observations echo those reported at other low-viscosity lava domes[12,21,24], and remind magmatic processes operating in the upper magma column of more basaltic open-systems[41]. Such multiparametric analyses could in the future be applied to other open-systems, and help constrain the overarching characteristics of open-vent volcanic activity and associated hazards[41,42].

## Methods

### Speckle filtering (SAR)

Speckle results from the interference of radar waves reflected by many elementary scatterers on the topography. It appears as granular noise in the SAR intensity images, which hinders their interpretation and reduces the ability to visualize small morphological features. To alleviate this problem, we have specifically designed and trained a convolutional neural network to remove speckle from SAR imagery acquired in volcanic settings, thereby gaining exceptional insights in fine structural details of the dome and crater (Supplementary Figs. 1 and 2). Importantly, the filter makes the analysis of the SAR images much more robust, as the noise suppression allows for an efficient recovery of the SAR shadow cast by small morphological aspects, otherwise very difficult in raw images. TSX images acquired over Colima volcano (Mexico) were used during training, and images acquired over Popocatépetl were used for testing exclusively. The filter is based on the Noise2Noise approach[61], whereby image denoising is learned without requiring noise-free reference data. The network is an adaptation and improvement of the filter designed for Sentinel-1 SAR images acquired in volcanic settings[62], which is used operationally by the volcano monitoring system MOUNTS[43]. The most important modifications were in the preprocessing and selection of the training data: since higher resolution requires more precise alignment, TSX crops were aligned individually, and discarded when alignment failed (large residual error due to morphological changes between the compared crops). The architecture is shown in Supplementary Fig. 3, along with details regarding the training procedure. An implementation for application with pretrained weights is available on GitHub (see "Code Availability").

### Crater depth, diameter, and volume estimations (SAR)

TSX SAR images were used to estimate crater depth and diameter. The images are analyzed in radar coordinates, in order to avoid geocoding warping artifacts, and to take advantage of the radar viewing geometry to recover crater dimensions. A total of 158 descending track acquisitions were analyzed from January 2012 to December 2019, from which 127 were acquired in spotlight-mode (range pixel spacing = 0.91 m, azimuth pixel spacing = 1.27 m), and 31 in stripmap-mode (range pixel spacing = 1.36 m, azimuth pixel spacing = 1.86 m), with an incidence angle of 44°. The orbital repeat time of the satellite is 11 days, however, the acquisition occasionally suffered interruptions, so that 78.3% of the

acquisitions had 11-day time interval, 11.5% 22-day interval, and the remaining ~10% had between 33- and 120-day interval (Supplementary Fig. 7b). The intensity images (horizontal polarization HH) were first despeckled using the trained convolutional neural network. The stack of all intensity images in radar geometry were then aligned and resized with respect to the first image, and cropped around the crater region. The images were then normalized, and successively binarized with a fixed intensity threshold, so that pixels with values below the threshold were considered as radar shadow regions (i.e., regions not reached by the radar beam). We chose a fixed intensity threshold of 0.25, and show the sensitivity to this threshold in Supplementary Fig. 4. Crater depth is estimated by measuring the length of the shadow cast by the inner-crater wall and multiplying it by the cosine of the radar incidence angle[29,31]. However, in many cases the crater had a more complex morphology (e.g., funnel-shaped crater, nested pit crater), which implied that the shadow was not well defined but rather a succession of discontinued shadows. For this reason, in order to recover the maximum crater depth $h_{crater}$, we counted the maximum number of "shadow" pixels encountered across a sequence of horizontal profiles crossing the crater, and multiplied it by the cosine of the radar incidence angle (Supplementary Fig. 4). The depth recovered in each horizontal profile was used to create the box plot in Fig. 2b, where boxes extend from the Q1 to Q3 quartile values of the calculated depths (with a line at the median Q2), and whiskers extend from the edges of box to show the range of values (whisker position set at 1.5·(Q3−Q1)). Recovering the depth from Sentinel-1 images (range pixel spacing = 2.33 m, azimuth pixel spacing = 13.99 m) was done in a similar way, the only difference being the image binarization method used to identify shadow regions in the image: instead of a fixed intensity threshold, we applied a graph cuts segmentation[63] on the logarithm of the raw intensity image, which was successively cleaned using a morphological operation (dilation with a cross-shaped operator). The reference point for the inner-crater calculated depth $h_{crater} = 0$ m in both TSX and S1 images is the main crater floor, so the progressive infilling of the main crater (due to progressive accumulation of pyroclastic deposits) is not affecting the measure. The main crater infilling (Supplementary Fig. 8a) was calculated from the radar shadow cast by the outer crater walls onto the main crater floor, south of the inner-crater. This region was chosen as representative of the average infilling rate of the terrace, however, because infilling could vary significantly from one region to another, we gave an error estimation based on the difference to the mean value measured in various points. The inner-crater diameter on the other hand was calculated by first applying a Sobel filter on the image, and secondly detecting the crater edges in the azimuth direction (i.e., vertical profile across the image), in order to avoid the foreshortening/layover effects inherent to SAR viewing in slant range. Images were checked and corrected manually when edges were not properly detected. The validation of the methods used to recover crater depth and diameter was achieved using a high-resolution digital elevation model, and proved that the values are recovered with a >98% accuracy (Supplementary Figs. 11 and 12). Considering the range pixel resolution and incidence angles of TSX and S1 products, the recovered depth vertical resolution is, respectively, of 0.65 m and 1.8 m. The combination of the crater radius and depth allowed to estimate the crater volume $V_c$, by assuming an inverted truncated cone geometry, with a minor base calculated to have crater slopes of 60°. Slopes of 40° and 80° were tested to give a error estimation on the recovered crater volume and extrusion volumes.

### Volcanic Radiative Power estimation (MODIS-MIROVA)

Thermal emissions were processed through the MIROVA system[64], an automated volcanic hotspot detection system based on the analysis of the MODIS sensor on board the Terra and Aqua satellites. The system calculates the Volcanic Radiative Power (VRP), which is a measurement of the heat flux radiated by hot volcanic surfaces expressed in Watt,

with a ±30% error on the measurement[64]. The combination of both satellites provide ~4 images/day of the entire Earth surface since 2000 and 2002, respectively, with a nominal spatial resolution of 1 km²/pixel in the infrared band.

## SO₂ degassing estimation (OMI)

SO₂ gas emissions were estimated from the analysis of the OMI sensor acquisitions on board the Aura satellite. The sensor measures solar backscatter radiation at wavelengths spanning from the visible to ultraviolet (270–500 nm), and provides daily global coverage since 2004, with nominal pixel spatial resolution of 13 × 24 km at nadir. The monthly SO₂ fluxes were calculated from the monthly SO₂ masses calibrated with the traverses method[65] (±50% error). The OMI images of each month were gridded and stacked over a 0.05° grid, excluding data affected by a thick cloud cover, or by the row anomaly (http://omi.fmi.fi/anomaly.html). The monthly cumulated SO₂ matrix was then divided by a matrix containing the number of valid data over the month, to obtain a map of the monthly averaged SO₂ around the volcano. A box was defined around the volcano to contain the monthly-averaged SO₂ anomaly corresponding to the volcanic plumes, and the monthly averaged SO₂ mass was calculated as the sum of the SO₂ column density of every grid element multiplied by its area. The calibration to convert the monthly averaged SO₂ mass into a flux, is obtained by performing a linear regression between a series of monthly masses (MM) and its corresponding series of monthly-averaged fluxes (MF) computed with the traverse method[65]. The regression plot between the two series obtained over 55 months at Popocatépetl is shown in Supplementary Fig. 13, with a best-fit ($R^2 = 0.759$) given by MF = 0.751·MM. In the case of Popocatépetl, a seasonal effect is noted in the monthly mass time series, which was corrected by multiplying the time series with a square sinusoidal function fitted to smooth this 12-month periodic fluctuation.

## Magma budget estimation

The magma budget of Popocatépetl was estimated following the approach of ref. 21. Volumes were calculated for Dense Rock Equivalent (DRE), assuming a DRE magma density $\rho_{DRE} = 2400$ kg/m³ [40], and an extruded lava (dome) bulk density $\rho_{extruded} = 2000$ kg/m³. The volume of degassed magma $V_{degas}$ is derived from the SO₂ OMI measurements using the "petrological method"[37]: $V_{degas} = MSO_2 / (2 \cdot \rho_{DRE} \cdot \Delta X_S)$, where MSO₂ is the measured mass of SO₂ in kg, and $\Delta X_S$ is the sulfur volatile loss. We used $\Delta X_S = 1500$ ppm, and tested two extreme cases $\Delta X_S = 500$ ppm (upper boundary for andesitic melt), and $\Delta X_S = 2500$ ppm (mafic melt mixed with andesitic melt). The volume of radiating magma $V_{thermal}$ is derived from the MODIS measurements using the "thermal approach"[66]: $V_{thermal} = VRE / c_{rad} \cdot (\rho_{extruded}/\rho_{DRE})$, where VRE (in J) is the volcanic radiant energy obtained from the integration of the VRP time series, and $c_{rad}$ (in J/m³) is an empirical coefficient accounting for the rheology of the extruding lava. We used a value $c_{rad} = 1.2 \times 10^7$ J/m³ consistent with andesitic lava domes, and tested two extreme cases $c_{rad} = 0.6 \times 10^7$ J/m³ and $c_{rad} = 1.8 \times 10^7$ J/m³. The volume of extruded magma $V_{extruded}$ is derived from the crater volume $V_c$, which was calculated from the TSX intensity images (see dedicated section in "Methods"). More specifically, the extruded DRE volume was taken as $V_{extruded} = \{\Delta V_c > 0\} \cdot (\rho_{dome}/\rho_{DRE})$, where $\Delta V_c$ is the discrete crater volume difference computed from one acquisition to another. We used a value $\rho_{extruded} = 2000$ kg/m³, and tested the extreme cases 1500 kg/m³ and 2500 kg/m³. The extruded volumes calculated here should be considered as a proxy for the average extruded magma volume, which are partially biased by the crater shape which we assume to be a truncated cone with slope angle of 60°. The volume of tephra $V_{tephra}$ expelled outside the crater is calculated from the semi-empirical equation $V_{tephra} = 1/3 \cdot V_{extruded}$. The factor 1/3 corresponds to the ratio between the extruded volumes of three distinct domes and the volume of tephra fall associated to their destruction (on 30 April 1996, 28

October 1996, and 30 June 1997), as reported by ref. 34 and ref. 67, respectively. This value is in agreement with the ratio of the cumulative volume of extruded lava to the cumulative volume of in-crater loss, shown by ref. 34 for the period 1996–2015. Although this factor is likely to vary significantly through time, it is an approximation used to give an order of magnitude to the tephra emissions $V_{tephra}$.

Magma fluxes $Q_{degas}$, $Q_{thermal}$, $Q_{extruded}$, and $Q_{tephra}$ are then derived from the corresponding volumes, by smoothing values using a 365-day running time-window, and averaging over monthly time bins.

## Modeling of crater deepening

The short-term fast subsidence observed after dome emplacements are explained by rapid variations in the volume fraction of exsolved gas in the upper magma column. We consider a simplistic approach where a cylindrical portion of the column of length $H_f$ and radius $R$ is filled with magma and exsolved gases, with an initial volumetric gas fraction $X_{gas0}$. Decreasing gas fraction in $H_f$ results in increasing column bulk density and decreasing column volume. The associated magma level variation $\Delta H$ can be recovered from $\Delta H = H - H_f$, where $H = (m_f/\rho_{DRE}(1-X_{gas}))/\pi R^2$. If we assume the mass of gas to be negligible with respect to the mass of magma, and the mass of magma within the cylindrical portion to be constant, such that $m(t) = m_f = \rho_{DRE}(1-X_{gas0}) \cdot \pi R^2 H_f$, than after simplification $\Delta H = H_f(1-X_{gas0})/(1-X_{gas}) - H_f$. We take $X_{gas0} = 0.5$ and $H_f = 100$ m ± 75 m following ref. 36 at Popocatépetl, and assume for simplicity that the gas fraction in $H_f$ decreases linearly until reaching values typical of pyroclast porosities erupted during dome-forming eruptions (-15–20%)[56,57]. The calculated magma level variations $\Delta H$ are compared to a selection of dome subsidence rates measured from TSX images (Fig. 6e) during the first ~30 days following dome emplacement ($t = 0$). The good fit suggests that a decrease in gas fraction from 0.5 to 0.2 in the first 30 days (i.e., $X_{gas}(t) = -0.01(t) + X_{gas0}$, where time is expressed in days) can explain the selected subsidence rates. Assuming a magma density $\rho_{DRE} = 2400$ kg/m³ [40], this gas fraction decrease corresponds to a column density increase from 1200 kg/m³ to 1920 kg/m³. The selection criteria for the subsidence trends shown in Fig. 6e was that dome subsidence could be measured during ≥2 TSX acquisitions without interruption by new lava extrusion; the corresponding TSX images are provided in Supplementary Fig. 5.

The long-term crater deepening observed on the timescale of years is reproduced using the theoretical model proposed by ref. 59. The model predicts pressure changes $\Delta P(t)$ in the magma reservoir due to steady gas loss (i.e., constant degassing rate), and the associated magma level variations $\Delta H(t)$ in the magma column are calculated following $\Delta P(t)/(g\rho_{m,c})$, where $g$ is the acceleration due to gravity, and $\rho_{m,c}$ is the mean density of melt (i.e., bubble-free magma) in the column. These magma level variations are used as a proxy to the observed crater depth (Fig. 7d). We consider the model scenario (1) in ref. 59, in which degassing occurs at low pressures via magma convection in the upper conduit[37,38,40,68]. The key model assumptions and chosen input values are the following (see model geometry and complete list of parameters in Supplementary Table 1):

1. A magma reservoir with initial volume $V_r$ is connected to an open magma-filled cylindrical conduit of length $L$. The reservoir pressure is magmastatic, and the entire conduit is subject to the same pressure change as the reservoir. This reservoir is not connected to a deeper magma source, as the continuous decrease in SO₂ emission rates observed since 2012 suggests that there was no major replenishment of deep undegassed magma. Although the geometry of Popocatépetl plumbing system is poorly constrained, most studies exclude the presence of a large magma chamber in the shallow crust (ref. 69 and references therein). We here use a conduit of length $L = 10$ km following ref. 36, and an initial reservoir volume $V_{r0} = 2$ km³. Note that the length of the conduit influences the initial reservoir pressure, but not the pressure variation $\Delta P$. The system is

embedded in a medium with viscoelastic rheology characterized by a bulk modulus $k = 10^{10}$ Pa and effective viscosity $\mu = 10^{18}$ Pa s.

2. Magma convection in the conduit is parametrized following ref. [60], where convection is driven by the density difference between degassed and undegassed magma. This parametrization states that the mean gas flux is related to the volumetric magma upflow rate $Q$, which depends on the conduit radius $R$ and magma viscosity. Magma convection in the conduit has been considered a plausible assumption at Popocatépetl[40], and the long-term excess degassing and excess thermal radiation reported here provide additional credit to it. The density difference between the degassed melt density $\rho_1$ and the undegassed melt density $\rho_2$ in the conduit is defined as $\Delta\rho_{1,2} = 59$ kg/m$^3$ following ref. 40. Because the descending degassed melt is replaced by ascending undegassed melt, the mean conduit melt density $\rho_{m,c}$ is assumed constant, and defined as the average density between $\rho_1$ and $\rho_2$. The model assumes the gas mass in the conduit to be much smaller than the mass of incompressible melt (liquid and solid phase), and assumes the mean gas density in the conduit to be much smaller than the mean melt density. Complex bubble dynamics such as foam collapse[70], which can cause short-term magma level variations, are not considered by the model.

3. Magma degassing rate is held constant through time. The degassing rate is controlled by the magma upflow rate, as per the convection parametrization described above. We use values of $Q_{degas}$ (DRE magma input derived from the measured SO$_2$ fluxes, with assumptions on magma sulfur concentration and DRE density) as the magma upflow rate $Q$. Three distinct values are considered to compute the pressure variation $\Delta P$: high $Q = 10$ m$^3$/s (i.e., $Q_{degas}$ early 2013, Fig. [5]), low $Q = 1$ m$^3$/s (i.e., $Q_{degas}$ late 2019, Fig. [5]), and intermediate $Q = 7$ m$^3$/s (i.e., flux considered by ref. [40]). According to the convection parametrization of ref. [60] (which uses a Poiseuille constant $\xi = 0.064$ and experimental constant of the effective conduit radius $R^* = 0.6$), these values equate to conduit radii of 14.8, 8.3, and 13.5 m, respectively. Conduit radius is held constant in the model.

4. Further parametrization of the model follows the degassing scenario (1) proposed by ref. [40] at Popocatépetl: the parent melt is a mixture of silicic and mafic end-members (65 wt% dacite + 35 wt% basaltic-andesite + 25 vol.% crystals), having a non-degassed density $\rho_{nd} = 2400$ kg/m$^3$, a non-degassed viscosity $\mu_{nd} = 10^4$ Pa s, and containing $\alpha = 3$ wt% of dissolved H$_2$O prior to degassing. Separation of gas and magma occurs in the upper conduit, where $n_c = 2$ wt% of H$_2$O is exsolved. The remaining 1 wt% H$_2$O retained in the melt is equivalent to depth of ~600 m (~150 bar), above which rests a permeable lava cap. Degassing results in a density difference $\Delta\rho_{1,2} = 59$ kg/m$^3$ between the undegassed and partially degassed melt, and in a degassed melt viscosity $\mu_1 = 10^{5.3}$ Pa s. Sensitivity of the model to varying initial reservoir volume $V_{r0}$ and mass fraction of exsolved gas in the conduit $n_c$ is shown in Supplementary Figs. 10a and 10b, respectively.

## Data availability

The TerraSAR-X raw data are available from the German Aerospace Center (DLR) eoweb service, and were obtained in this study as part of the project TSX-ID 1505. TerraSAR-X spotlight-mode data were acquired through proposals twal_GEO1505 and bmc_GFZ_walter. TerraSAR-X despeckled images generated in this study are published as ref. [71] and archived at https://zenodo.org/record/7842336. Sentinel-1 raw data are freely available from the European Space Agency's (ESA) Copernicus Open Access Hub (https://scihub.copernicus.eu/), and filtered intensity images are visible on the MOUNTS platform (http://mounts-project.com/timeseries/341090). MODIS and OMI data are freely available from NASA's LANCE system (http://lance-modis.eosdis.nasa.gov/) and Goddard Earth Sciences Data and Information Services Center (GES DISC, https://disc.gsfc.nasa.gov/), respectively. Daily reports on the volcanic activity of Popocatépetl are available from the Centro Nacional de Prevención de Desastres (CENAPRED, https://www.cenapred.unam.mx/reportesVolcanGobMX/), and include surveillance camera images and videos. The data generated in this study are provided in the Supplementary Data 1 file.

## Code availability

The trained CNN used to filter speckle in TSX images is provided through a GitHub repository along with usage instructions: https://github.com/Andreas-Ley/S2S-TSX-Colima. The code version used in this study is published as ref. [72] and archived at https://zenodo.org/record/7838864.

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

## Acknowledgements
S.V. was supported by DGAPA-PAPIIT grant IA102221, T.R.W. by ERC consolidator grant ERC-CoG Q7 646858, R.C. by SEP-CONACYT grant A1-S-30127. We thank DLR for the continuous acquisition of TerraSAR-X spotlight-mode data, ESA and NASA for providing access to open-access data (Sentinel-1, MODIS, OMI), and CENAPRED for providing access to daily volcanic activity reports.

## Author contributions
S.V. designed the study, conducted the data analysis, wrote the manuscript, and created the figures. D.C., R.C., A.L., T.B., N.T., D.L., M.L., T.R.W., and S.C.-R. contributed to the development of ideas and concepts, and provided critical comments during manuscript preparation. D.C. and M.L. conducted the collection and analysis of MODIS data to recover volcanic radiative power, R.C. conducted the collection and analysis of OMI data to recover $SO_2$ fluxes, A.L. designed and trained the CNN for TSX speckle filtering, T.B. conducted analysis of Google Earth images.

## Competing interests
The authors declare no competing interests.
