## [Peer Review File · Nature Communications]

Reviewers' Comments:

Reviewer #1:

Remarks to the Author:

This study presents high-resolution satellite radar imagery of the construction-subsidence dome cycles at Popocatepetl volcano, which suggests that these cycles can be interpreted as gas-driven convection surges. The observations are derived from high temporal and spatial resolution data of the dome morphology and depth, SO₂ gas emission, and infrared thermal radiation observation over eight years that encompass several dome construction-destruction cycles. The comprehensive analysis of these data provides convincing arguments that the short-term dome construction-subsidence cycles can be explained in terms of gas-driven convection surges, whereas long-term inner-crater deepening is associated with the depressurization of the system due to gas depletion. Considering the relevance of this study to better understand the governing process of the repetitive dome cycles at Popocatepetl volcano and similar volcanoes, the high-quality data, sound methodology, and convincing interpretation, I am happy to recommend this manuscript for publication in Nature Communication after a minor revision.

General comments:

1) I appreciate the significance of the analysis presented in this study, however, I think that it is important to emphasize the originality of this comprehensive approach in the introduction and discussion sections to help a broader audience in related fields to better understand how this article represents a breakthrough in this topic with respect to previous studies (such as references 14, 26, 27, 41,42,50).

2) The relevance and applicability of this study should also be highlighted. For instance, you can briefly discuss its implications for short and long-term eruptive scenarios and their associated hazards at Popocatepetl volcano, in what kind of systems these results could be relevant, and associated research topics that can benefit from this study.

Minor comments:

Line 110. Remove "other".

Line 134. Reference 13 is more adequate than Ref. 41 for the strombolian-like activity (not mentioned in the latter).

Line 140. "the subsidence starts shortly after the dome emplacement (likely within the first hours-day). How do you know this timescale, considering that the TXS SAR images have 11-day time interval?"

Figure 4a is not referred to in the text.

Reviewer #2:

Remarks to the Author:

This paper presents an analysis of the evolution of the Popocatepetl volcanic dome, and integrates multiple datasets and approaches. The study is interesting and noteworthy, but I noted some issues on data analysis that I recommend to address or clarify:

Major comments:

- Speckle filtering (METHODS). This is a fundamental step in the work presented by the authors and, from my point of view, more details about training and machine learning architecture are needed to make the study easily reproducible by other researchers. Also, what is the sensitivity of your model to the hyperparameters of the neural network? How sensitive are your results to those hyperparameters? How do you ensure that the same model applies to another dome/volcano? How did you validate your model? Validation should be performed with labelled data to make sure that you can use the model from Colima on Popocatepetl. I understand that validation may be challenging in this problem, but a more comprehensive justification of the approach would be needed.

- Line 211/458-464. I'm not convinced with the use of this empirical equation. The factor 1/3 is estimated from three other domes only, and it is unclear to what extent the same factor can apply to the dome studied here. Also, what is the uncertainty associated to this factor? There appears to

be a lot of (unknown) uncertainty in that estimation, so I don't think the calculation of V_{tephra} is reliable. Also, why do the factor $1/3$ serves "as a conservative value which likely overestimates the actual erupted tephra volume V_{tephra} "? What is the argument that supports that statement?

- Line 379-381 and Fig. 1. Please revise this. Based on the scheme in your figure, and if I understand properly how you define the radar incidence angle (missing in the figure), the crater depth cannot be calculated multiplying the shadow by the cosine. You should use: $\tan I = \text{shadow/depth}$, where \tan is the tangent and I is the radar incidence angle. This may produce some significant changes in your results.

- Lines 492-494. Is this SO_2 flux or total (mostly water vapor) gas flux? Are the units correct? The model used here uses Q in kg/s , and refers to total gas flux, not to SO_2 flux only. How do you convert from kg/s to m^3/s ? To do that, you would need to assign a pressure value. What pressure do you assume for that conversion? Please, clarify.

- Line 378. How are your results affected by this arbitrary threshold? A sensitivity analysis would help to assess the robustness of your approach.

Minor comments:

- Lines 77-78. Improve transition from one paragraph to the other. Too sharp change of topic.
- Line 212. How is smoothing performed? More details needed.
- Line 237-237. The mention to seismic tremor here looks a bit out of place (it is not mentioned in any other place in the text). Why is important to mention tremor? If you want to add a mention to tremor, you should add references (many references to seismic tremor related to fluid flow inside conduits are missing).
- I suggest adding a "Conclusions" section.

Reviewer #3:

Remarks to the Author:

Review of the manuscript : Dome cycles and crater deepening at Popocatepetl reveal convection surges of a depressurizing magma column. By S. Valade et al.

The manuscript presents a study of the lava activity at Popocatepetl using SAR imaging, SO_2 and thermal images and deep-learning to filter the SAR images.

My opinion is mixed:

On the positive side, the quality of the SAR images is impressive and the method is a major contribution to dome monitoring. The movements of the lava column is very clear. The comparison between the different data sheds new light on the dynamics of Popocatepetl. The manuscript is well written and the illustrations are very clear. The science behind it seems very serious.

The negative side is perhaps related to the manuscript length and to the wish to place this research in a broader context: (1) one of my biggest reservations are related to the title and the introduction which focuses on lava domes. The case of Popocatepetl is very interesting but it does not appear to be a lava dome. The data presented show that the lava always remains below the bottom of the crater. It is rather the top of a magma conduit or a small viscous "lava lake". The issues are therefore very different from those of domes that grow at the top of volcanoes and threaten to collapse. The title and the introduction thus appear not adapted to the scientific content. (2) The much emphasised deep-learning aspect is a filter that enhances the visual aspect but - from the figures available - does not really seem to reveal aspects that would not be seen on the raw images. (3) Each section of the results and the discussion would deserve more explanations on the values chosen, the limitations of the models and the assumptions done. Even if the conclusions of the authors seem realistic, they also seem very speculative. What evidence for gas-driven convection surges? Why crater deepening would be linked to progressive gas depletion

and not to gravitational collapse when the magma column sinks? Why convection within the magma column and not below in a larger chamber? Is 40-70% bubbles (line 270) realistic in a magma that seems very fluid? Even with the Methods, these questions remain.

In conclusion, the scientific background is present but due to its current form, I am not sure that the article can be published in Nature Communication.

Response to Referees

Nature Communications manuscript NCOMMS-22-38331

Valade et al. "Dome cycles and crater deepening at Popocatépetl reveal convection surges of a depressurizing magma column"

We warmly thank all three reviewers for their valuable critics, which we have found very constructive and helpful. We have done our best to apply the suggested modifications, and now believe that the manuscript has been clearly improved. Below is a point-by-point reply to each reviewer, which we hope they will find satisfying.

1. Response to Reviewer #1

Reviewer #1 (Remarks to the Author):

This study presents high-resolution satellite radar imagery of the construction-subsidence dome cycles at Popocatépetl volcano, which suggests that these cycles can be interpreted as gas-driven convection surges. The observations are derived from high temporal and spatial resolution data of the dome morphology and depth, SO₂ gas emission, and infrared thermal radiation observation over eight years that encompass several dome construction-destruction cycles. The comprehensive analysis of these data provides convincing arguments that the short-term dome construction-subsidence cycles can be explained in terms of gas-driven convection surges, whereas long-term inner-crater deepening is associated with the depressurization of the system due to gas depletion.

Considering the relevance of this study to better understand the governing process of the repetitive dome cycles at Popocatépetl volcano and similar volcanoes, the high-quality data, sound methodology, and convincing interpretation, I am happy to recommend this manuscript for publication in Nature Communication after a minor revision.

We wish to thank the reviewer for the constructive comments which have helped improve the manuscript. We give below answers to each concern raised, and explain how the manuscript has been modified. Please note that the line numbers Lxxx used throughout our answers refer to the revised *track-change* manuscript, in order to easily visualize how the manuscript was edited.

General comments:

1) I appreciate the significance of the analysis presented in this study, however, I think that it is important to emphasize the originality of this comprehensive approach in the introduction and discussion sections to help a broader audience in related fields to better understand how

this article represents a breakthrough in this topic with respect to previous studies (such as references 14, 26, 27, 41,42,50).

Reply: We truly appreciate the comment. We have now modified the end of the introduction (L116-124), in order to stress that the data/methodology used here are unique, and offer a new comprehensive view of the processes operating at Popocatépetl. We now also emphasize the implications of this study for both the interpretation and monitoring of other open-vent systems volcanoes. Two new references (Vergniolle and Métrich 2022, Edmonds et al. 2022) have been added to emphasize that such multiparametric analysis could in the future be applied to other open-systems to help constrain the overarching characteristics of open-vent volcanic activity and associated hazards. Lastly, a new conclusion paragraph has also been added at the end of the discussions to summarize the outcomes and perspectives of our study.

Vergniolle, S. & Métrich, N. An interpretative view of open-vent volcanoes. Bulletin of Volcanology 84, 83 (2022).
Edmonds, M., Liu, E. J. & Cashman, K. V. Open-vent volcanoes fuelled by depth-integrated magma degassing. Bulletin of Volcanology 84, 1–27 (2022).

2) The relevance and applicability of this study should also be highlighted. For instance, you can briefly discuss its implications for short and long-term eruptive scenarios and their associated hazards at Popocatépetl volcano, in what kind of systems these results could be relevant, and associated research topics that can benefit from this study.

Reply: Thank you for pointing this out. As answered above, the introduction has been changed to emphasize these aspects. We also now added a conclusion paragraph at the end of the discussions, in order to (1) stress the novelty of these combined observations, (2) summarize the main outcomes of the research, and (3) stress that the methods can be applied at other open-volcanoes (where similar observations are made), with emphasis on the fact that the study paves the way to improved multidisciplinary satellite volcano monitoring with obvious benefits for hazard assessment.

Minor comments:

- Line 110. Remove “other”.

Reply: The term “other” is meant to explicitly express that the observations reported here are observed at other volcanoes (i.e., Lásca, Mount Cleveland), so we feel it is important to keep this word.

- Line 134. Reference 13 is more adequate than Ref. 41 for the strombolian-like activity (not mentioned in the latter).

Reply: Done

- Line 140. “the subsidence starts shortly after the dome emplacement (likely within the first hours-day). How do you know this timescale, considering that the TXS SAR images have 11-day time interval?”

Reply: This is a very relevant observation which indeed needs clarification. If we take the April 2016 dome cycle presented in Figure 1, the CENAPRED reports the end of the

strombolian activity at ~11:00 GMT on 2016-04-18 (see report on 18 April 2016 at <https://www.cenapred.unam.mx/reportesVolcanGobMX/>). The next day 2016-04-19, CENAPRED only reports “intense incandescence above the crater observed during the night”. The first TSX image acquired after this episode is on 2016-04-20 12:28 GMT, hence ~2 days after the end of the strombolian activity, and shows the first signs of dome subsidence (i.e., central pit and peripheral bulge, see Figure 1b). So from a satellite point of view, we can assume that the subsidence process starts within the first 2 days following its emplacement. We now have updated the text accordingly L164: “The subsidence starts shortly after the dome emplacement (within the first 2 days according to Fig. 1i and 1b)”.

- Figure 4a is not referred to in the text.

Reply: The reference has now been added to the text at L218.

2. Response to Reviewer #2

Reviewer #2 (Remarks to the Author):

We wish to thank the reviewer for the constructive comments which have helped improve the manuscript. We give below answers to each concern raised, and explain how the manuscript has been modified. Please note that the line numbers Lxxx used throughout our answers refer to the revised *track-change* manuscript, in order to easily visualize how the manuscript was edited.

This paper presents an analysis of the evolution of the Popocatepetl volcanic dome, and integrates multiple datasets and approaches. The study is interesting and noteworthy, but I noted some issues on data analysis that I recommend to address or clarify:

Major comments:

- Speckle filtering (METHODS). This is a fundamental step in the work presented by the authors and, from my point of view, more details about training and machine learning architecture are needed to make the study easily reproducible by other researchers. Also, what is the sensitivity of your model to the hyperparameters of the neural network? How sensitive are your results to those hyperparameters? How do you ensure that the same model applies to another dome/volcano? How did you validate your model? Validation should be performed with labelled data to make sure that you can use the model from Colima on Popocatepetl. I understand that validation may be challenging in this problem, but a more comprehensive justification of the approach would be needed.

Reply: The reviewer is correct, we had chosen to leave out the details of the filter development thinking that it was too technical. Responding to the reviewer's comment, we now have added further details on the speckle filtering: the new supplementary figure 3 shows the chosen network architecture, as well as details regarding the training procedure. Also, we shared the pretrained inference code on GitHub to facilitate reproduction, so that other researchers can apply the filter to their dataset (<https://github.com/Andreas-Ley/S2S-TSX-Colima>). The Methods section "Speckle filtering (SAR)" has been modified to refer to this additional material.

Regarding the hyper parameters: the network is not sensitive to the choice of hyper parameters, which makes the training quite stable. However, the training data preparation involves a couple of non-obvious steps. To this end, we have given details on the criteria used for image pair selection in the Supp. Fig. 3.

Regarding the model validation: model validation is difficult as ground truth data for speckle filtering in this context is not readily available. Indeed, the usual approach to speckle filtering

is to consider pairs of noise-free and speckled images in a regression framework, but speckle-free scenes are not available and need to be artificially created. Some authors (Wang et al. 2017) have generated training data by adding speckle to optical data, however due to the acquisition geometry of SAR and the complexity of target models, this training data may not be consistent with real SAR images. As described in the Methods section of the manuscript, we here circumvent this problem using the “Noise2Noise” approach (Lehtinen et al. 2018), which does not require speckle-free images since it uses the redundancy of reflectivity information in multiple SAR images acquired over the same region.

In order to give the reader some additional information to evaluate qualitatively the performance of the filter, we have added a new supplementary figure (now Supp. Fig. 1), which compares crops of both raw and filtered TSX images acquired at Popocatépetl volcano. We stress that it corresponds to a region that was not used during training (which used images from Colima only), and therefore demonstrates qualitatively the efficiency of the filter. As mentioned above, the reader now also has access to the filter (shared through a GitHub repository) to test on his/her own dataset.

Wang, P. , Zhang, H. and Patel, V. M. (2017) “Sar image despeckling using a convolutional neural network,” IEEE Signal Processing Letters, vol. 24, no. 12, pp. 1763–1767

Lehtinen, J. et al. Noise2Noise: Learning image restoration without clean data. in 35th International Conference on Machine Learning, ICML 2018 vol. 7 4620–4631 (2018)

- Line 211/458-464. I'm not convinced with the use of this empirical equation. The factor 1/3 is estimated from three other domes only, and it is unclear to what extent the same factor can apply to the dome studied here. Also, what is the uncertainty associated to this factor? There appears to be a lot of (unknown) uncertainty in that estimation, so I don't think the calculation of V_{tephra} is reliable. Also, why do the factor 1/3 serves “as a conservative value which likely overestimates the actual erupted tephra volume V_{tephra} ”? What is the argument that supports that statement?

Reply: To support the empirical equation, one may look at Figure 12 (see below) of the cited reference Gómez-Vazquez et al. 2016, which plots the cumulative volume of extruded lava (black squares), and the cumulative volume of in-crater loss (blue circles) for the period 1996-2015. Looking for example at the condition in say 2003, the accumulated volume of lava is $30 \times 10^6 \text{ m}^3$, while the total material loss within the crater is $20 \times 10^6 \text{ m}^3$. This means that a volume $10 \times 10^6 \text{ m}^3$ is “missing”, that is, it was expelled outside the crater as tephra by explosions. The ratio of the total expelled material to the cumulative lava extruded is thus about 1/3. This has been made explicit in both the main text L236 and Methods L548. Also, the statement saying that it “likely overestimates the actual erupted tephra” has been replaced by saying that it is “used to give an order of magnitude to the tephra emissions”.

Figure 12 in Gómez-Vazquez et al. 2016 (ref. 38)

- Line 379-381 and Fig. 1. Please revise this. Based on the scheme in your figure, and if I understand properly how you define the radar incidence angle (missing in the figure), the crater depth cannot be calculated multiplying the shadow by the cosine. You should use: $\tan I = \text{shadow}/\text{depth}$, where \tan is the tangent and I is the radar incidence angle. This may produce some significant changes in your results.

Reply: We have improved the sketch in Fig. 1 by adding the radar incidence angle, and by changing the position of the shadow which was misleading. Indeed, distances measured along the x-axis of SAR images correspond to distances along the radar line-of-sight (LOS), i.e. along the beam propagation direction (range direction). The shadow width (W) thus corresponds to the hypotenuse of the right-angled triangle, and the crater depth (H) is the side adjacent to the radar beam incidence angle (θ). The formulation is thus: $H = W \cdot \cos(\theta) \cdot \text{LOS_pixel_spacing}$, as described in the methods and applied in the manuscript.

- Lines 492-494. Is this SO₂ flux or total (mostly water vapor) gas flux? Are the units correct? The model used here uses Q in kg/s, and refers to total gas flux, not to SO₂ flux only. How do you convert from kg/s to m³/s? To do that, you would need to assign a pressure value. What pressure do you assume for that conversion? Please, clarify.

Reply: This is actually a DRE magma input flux, which is itself recovered from the SO₂ flux: thank you for pointing this out! The text has been corrected to avoid any confusion at L347, L578 and L600, as well as Figure 7 & Supp. Table 1.

The units are correct, but indeed needed clarification. The model used here (Girona et al. 2014, ref. 57) uses the scenario in which degassing is coupled with magma convection in the conduit, as parameterized by Stevenson and Blake 1998 (ref. 58). The latter state that “gas flux depends on the volumetric magma upflow rate (Q)”, which is expressed in m³/s. In Girona et al. 2014, although the “mean gas flux” is expressed in kt/d in the Figures, the equations implement the exact parametrization of Stevenson and Blake 1998, where Q is expressed in m³/s (and the parameter α defines the mass fraction of volatiles dissolved in the parent melt, and n_c defines the mass fraction of exsolved volatiles from the conduit). Our approach is therefore the following: from the measured SO₂ flux [kg/s] we compute a DRE magma input

rate [m^3/s] (i.e., Q_{degas} in Fig. 5 ranging from 1-10 m^3/s), by assuming a range of sulfur concentration in the magma [ppm] and a magma DRE density (see Methods section “Magma budget estimation”), which we then feed to the model of Girona et al. 2014.

- Line 378. How are your results affected by this arbitrary threshold? A sensitivity analysis would help to assess the robustness of your approach.

Reply: The threshold value was selected “manually” on the basis of a representative sample of images, with an attempt to detect shadow regions only in the SAR images. However a sensitivity analysis on this threshold was indeed lacking from the manuscript. We have consequently performed this analysis, and have now included this as an additional Supplementary Figure (now Supp. Fig. 4). The text in the Methods section has been slightly expanded with reference to this additional figure (L452, L460).

Minor comments:

- Lines 77-78. Improve transition from one paragraph to the other. Too sharp change of topic.

Reply: Done, a sentence has been added to improve the transition L77.

- Line 212. How is smoothing performed? More details needed.

Reply: Good point, this information was added to the Methods section L554, and reference to the methods was added to the text L239.

- Line 237-237. The mention to seismic tremor here looks a bit out of place (it is not mentioned in any other place in the text). Why is important to mention tremor? If you want to add a mention to tremor, you should add references (many references to seismic tremor related to fluid flow inside conduits are missing).

Reply: It seemed important to mention tremor because it is one of the indicators of fluid flow within the conduit. We have therefore rewritten that line as: “and iv) fluid flow inside the conduit revealed by characteristic seismic tremor signal^{38,49}”, and added a new reference (Arciniega-Ceballos et al. 2003) to support this claim:

Arciniega-Ceballos A., Chouet B., Dawson P. Long-period events and tremor at Popocatepetl volcano (1994–2000) and their broadband characteristics. Bulletin of Volcanology 65,124-35 (2003)

- I suggest adding a “Conclusions” section.

Reply: We now added a conclusion paragraph at the end of the discussions, in order to (1) stress the novelty of these combined observations, (2) summarize the main outcomes of the research, and (3) stress that the methods can be applied at other open-volcanoes, where similar observations are made. Please note that this paragraph is not under an explicit “conclusions” section, as it appears to us that in the vast majority of cases, Nature Communications are organized with the standard “Introduction, Results, Discussion and Methods” sections. Nevertheless, we leave the final decision to the Editor as to whether an explicit “conclusion” section should be created or not.

3. Response to Reviewer #3

Reviewer #3 (Remarks to the Author):

We wish to thank the reviewer for his time, and for his critical comments which have helped mature and clarify various aspects of the manuscript. We give below answers to each concern raised, and explain how the manuscript has been modified, or in the contrary, why it has been left unchanged. Please note that the line numbers Lxxx used throughout our answers refer to the revised *track-change* manuscript, in order to easily visualize how the manuscript was edited.

The manuscript presents a study of the lava activity at Popocatépetl using SAR imaging, SO₂ and thermal images and deep-learning to filter the SAR images.

My opinion is mixed:

On the positive side, the quality of the SAR images is impressive and the method is a major contribution to dome monitoring. The movements of the lava column is very clear. The comparison between the different data sheds new light on the dynamics of Popocatepetl. The manuscript is well written and the illustrations are very clear. The science behind it seems very serious.

The negative side is perhaps related to the manuscript length and to the wish to place this research in a broader context:

(1) one of my biggest reservations are related to the title and the introduction which focuses on lava domes. The case of Popocatepetl is very interesting but it does not appear to be a lava dome. The data presented show that the lava always remains below the bottom of the crater. It is rather the top of a magma conduit or a small viscous “lava lake”. The issues are therefore very different from those of domes that grow at the top of volcanoes and threaten to collapse. The title and the introduction thus appear not adapted to the scientific content.

Reply: The point raised by the reviewer is in fact very relevant, and called for some clarifications.

On one hand, let us start by agreeing on the fact that what we are indeed observing the top of a magma conduit, and that the so-called “dome growth-subsidence cycles” are reflecting the “ups and downs of the upper lava column”. We have now tried to make this even more explicit throughout the manuscript, in particular at L96, L118, L277, L292, L360. Also, in order to clarify our definition of what we call “inner crater”, and to clarify why we use “lava domes” in our paper, we have inserted a several sentences in the introduction (L85-97), which give a broader historical context of Popocatépetl recent eruptive history to support these definitions.

On the other hand, we actually believe that the term “lava dome” (as defined in the first lines of the introduction with the corresponding references) is appropriate at Popocatépetl (as justified L82-85), and that we should keep this terminology throughout the manuscript. We here provide some arguments for that, on the basis of the concerns raised by the reviewer, and state how we modified the manuscript to convince the reviewer and reader:

- *“It is rather the top of a magma conduit or a small viscous “lava lake””*

Reply: as mentioned above we agree that we are observing the top of a magma conduit. However the term “lava lake” is not appropriate in our opinion, as it would imply surface movement indicating internal convection (even in very viscous phonolite lava lakes such as Erebus), which is not the case.

- *“The issues are very different from those of domes that grow at the top of volcanoes and threaten to collapse”*

Reply: The reviewer here likely refers to highly viscous domes, e.g., Peleean domes. However according Blake 1990 in particular, this is only one hand of the spectrum (as mentioned L41-43). The other hand of the spectrum are the “low domes”, of which Popocatépetl is a case example. Regarding the “issues”, it is true that the main dome hazard at Popocatépetl is not related to a “collapse” stricto sensu (i.e., gravitational collapse of a dome portion, such as Unzen 1991). This is however partly related to the current crater morphology, and it is not excluded that if the domes were to reach the flanks, that these would not generate the so-called coulées, potentially subject to pyroclastic flows mentioned in the introduction L50. Importantly, we sustain that the “associated volcanic hazards of [...] domes are highly dependent on the [...] dome morphology”, even in the case of low domes such as Popocatépetl. Indeed, as described L51 and L281, we believe that the subsidence process leads to “rapid changes in the dome permeability” which “lead to sudden transitions from passive degassing to violent explosive events”. As such, we chose to keep this part of the introduction related to the dome hazards unchanged. We have however added a reference (Hyman et al. 2018) addressing the role of pressure distribution in domes at Popocatépetl from numerical modeling.

Hyman, D. M., Bursik, M. I. & Legorreta Paulín, G. Time Dependence of Passive Degassing at Volcán Popocatépetl, Mexico, From Infrared Measurements: Implications for Gas Pressure Distribution and Lava Dome Stability. Journal of Geophysical Research: Solid Earth 123, 8527–8547 (2018).

- *“The data presented show that the lava always remains below the bottom of the crater”*

Reply: it is very true that from ~2013 onwards, the inner-crater deepens significantly, and that the lava remains at the bottom of this crater. Nonetheless, the domes observed early 2012 are actually emplaced at a level close to that of the main crater (as mentioned L188), and have the typical “low dome” morphology defined by Ref. 3. (Blake, S. *Viscoplastic Models of Lava Domes*, 1990). The new sentences added L85-97), now clarify our definition of what we call “inner crater”, and clarify why we use the term “lava domes” at Popocatépetl.

- *“One of my biggest reservations are related to the title and the introduction which focuses on lava domes”*

Reply: At the time of submission, we seriously considered the following title for our manuscript: *“Imaging the cyclic ups and downs of a magma conduit at Popocatepetl volcano, Mexico”*. Based on the reviewer’s comments, we imagine that he/she might have preferred it with respect to the current title. Although we found it very attractive (and continue to do so), we intentionally decided to keep a title explicitly referring to lava domes. Indeed, all literature on Popocatepetl’s current activity refers to “lava domes”, and we thought it wise to mention this word in the title to bring broad attention to the paper. Secondly, it allowed us to explicitly introduce in the title that the crater is deepening. Despite not changing the title in the revised manuscript, and because we fully agree that the observed dome growth-subsidence cycles reflect the “ups and downs of the upper lava column”, we have added explicit mentions of this throughout the manuscript (as mentioned earlier).

(2) The much emphasised deep-learning aspect is a filter that enhances the visual aspect but - from the figures available - does not really seem to reveal aspects that would not be seen on the raw images.

Reply: It is true that the filter does not actually reveal features that are not seen in raw images. It does however significantly ease the visualization of small morphological features, which otherwise are hardly visible due to the speckle “noise” (e.g., fine ring fractures, ballistic blocs, etc.). We consequently edited the text to tune down the claim at L142 which was too bold. Importantly, the filter makes the analysis of the images much more robust. Indeed, the noise suppression allows for an efficient recovery of the SAR shadow cast by small morphological aspects, which is otherwise near impossible. This has been explicitly mentioned by adding a sentence in the Methods L421. We have also added a new supplementary figure (now Supp. Fig. 1), which shows various crops of the raw and filtered amplitude image, and helps the reader better visualize the performance of the filter. We stress that the displayed image corresponds to a region that was not used during training (which used images from Colima only), and therefore demonstrates qualitatively the efficiency of the filter. The Supplementary Figure 2 has been left unchanged, and shows the reader how the filtered images allows enhanced visualization of the dome morphological evolution with respect to raw images (for both high-resolution TSX images and low-resolution Sentinel-1 images), and how it was fundamental for the precise recovery of crater depths.

Lastly, we stress that we have provided further details on the filter itself (architecture and training procedure) by adding a new supplementary figure (now Supp. Fig. 3). Last but not least, we now shared the filter’s pretrained inference code on GitHub (<https://github.com/Andreas-Ley/S2S-TSX-Colima>), so the reader can now apply it to his/her own dataset.

(3) Each section of the results and the discussion would deserve more explanations on the values chosen, the limitations of the models and the assumptions done. Even if the conclusions of the authors seem realistic, they also seem very speculative.

Reply: We have tried to give more explanations in various sections of the Results and Discussions, by providing error bars where possible, adding Supplementary Figures when needed, and inserting additional comments in the manuscript when it felt necessary.

We have in particular addressed the following specific points in the Results section:

- inner-crater slope assumption: the assumption of using slopes of 60° (based on Fig. 2B in Macías et al. 2020 JVGR, where slopes are “greater than 56°”) had an influence on the values cited in the text, and in particular of the inner-crater volume loss and the extruded dome volumes. In order to show the influence of such a geometrical assumption, we have also computed the values using slopes of 40° and 80° (as now mentioned at L196, L488), and edited the text by providing error range regarding extruded dome volumes L156, and crater volume loss L195. We have also edited Fig. 3b by adding a gray envelope reflecting this.

Macías, J.L. et al. (2020), Source and behavior of pyroclastic density currents generated by Vulcanian-style explosions of Popocatepetl volcano (Mexico) on 22 January 2001, Journal of Volcanology and Geothermal Research, 406, 107071, [10.1016/j.jvolgeores.2020.107071](https://doi.org/10.1016/j.jvolgeores.2020.107071)

- main-crater infilling volume: we have added an error bar on this value L201 and on the resulting infilling volume L204, and detailed how it was recovered in the methods L474.
- assumed value 1/3 to compute volume of tephra: it might have appeared too trivial to compute this value on 3 domes only. We have modified the text in both the results and methods, to stress that this value is in agreement with the ratio of the cumulative volume of extruded lava to the cumulative volume of in-crater loss shown by Gómez-Vazquez et al. 2016 (ref. 37) for the period 1996-2015. Also, we now stress L552 that although this factor is likely to change through time, it is here used to give an order of magnitude to the tephra emissions.
- VRP thermal radiation and SO₂ flux measurement error: we have specified the error on the measurements in the Methods section, referring to the appropriate literature for further information.
- SAR depth measurement resolution: we have added details regarding the depth resolution for both TSX and S1 measurements in the Methods section L484.
- SAR depth sensitivity to fixed threshold: a new supplementary figure was added (now Supp. Fig. 4) to show the sensitivity of the recovered depth to the fixed intensity threshold

In the Discussion section on the other hand we have tried to stress more the limitations and assumptions of the following points:

- Long-term crater deepening model: we acknowledge that the model tested here (i.e., whereby persistent degassing leads to system depressurization and progressive crater deepening/widening) was lacking explanation on the chosen values and underlying assumptions. We have now added several sentences in the Methods section “Modeling of crater deepening” L571-599 to explain some key assumptions of the model, and importantly, have explained the chosen parametrization and associated values (mainly taken from the degassing scenario presented by Witter et al. 2005 at

Popocatepetl). A sentence has now also been added in the discussion L339. The parameter values (and associated reference) are listed in the Supplementary Table 1. The Supplementary Figure 10 (formerly Supp. Fig. 7) has been slightly modified to show the influence of the exsolved gas fraction n_c (rather than the influence of the magma conduit length L as it was previously done). Indeed, the conduit length influences the initial reservoir pressure (which is mainly magmastic), but has no influence on the calculated pressure variation ΔP . Of course the reader is referred to the original paper (Girona et al. 2014) which describes the model in great detail and shows further parametric analysis which extend the ones presented here.

Lastly, we wanted to stress that although this model seems able to explain several observations (i.e. crater deepening and expected conduit radius) when using reasonable input values taken from both the literature and our study, it still remains speculative. We therefore have added a small paragraph at the end of the discussion section L388-393 to acknowledge that, and to discuss the implications of this tested model. In particular, we answer the reviewer's next question (*"Why crater deepening would be linked to progressive gas depletion and not to gravitational collapse when the magma column sinks?"*).

- Evidence for shallow magma convection: as explained below in the reviewer's question on *"Why convection within the magma column and not below in a larger chamber?"*, we added a couple of sentences at the end of the discussion section L388-393 to discuss the limits of our conclusions and the future work to carry out to constrain this aspect.

What evidence for gas-driven convection surges?

Reply: The most straightforward mechanism to drive magma upwards is to increase its buoyancy through increase of its gas fraction. This seems in agreement with the reported increase in SO_2 emissions and intense strombolian activity witnessed during the dome emplacement (as mentioned L251, along with the supporting references). On the other hand, we use the term "convection" to illustrate the upward and successively downward advection of magma, which characterize the observed dome growth-subsidence cycles. Although this seems counter-intuitive, we have tried to explain and model in the section *"Domes as convection pulses"* that gas retention and escape from the upper magma column can explain our observations. The term "surge" is associated to the term convection, to indicate the rapid nature of the phenomena (at least the dome emplacement phase, Fig. 1).

Why crater deepening would be linked to progressive gas depletion and not to gravitational collapse when the magma column sinks?

Reply: This is an interesting point. In fact, we believe that these two processes are not contradictory, but rather interdependent. This has been explicitly mentioned in the discussions L359-362. Indeed, in our opinion the long-term gas depletion of the magma column induces its densification, which consequently sinks, and results in the gravitational collapse of the crater. Conceptually, we had tried to picture this aspect in Fig. 6b, but it was clearly not

enough. From a physical point of view, we should stress that the model expresses this relationship explicitly, as the magma level variation ΔH is calculated from the magma column pressure variation ΔP (see Methods). We also stress that the mechanism behind this model (long-term crater deepening, Fig 7b) is similar to the model for the short-term dome/crater deepening, which relates the decrease in the gas fraction in the upper column, to a decrease of the density, and consequently a drop of the column ΔH (Fig 7a).

Why convection within the magma column and not below in a larger chamber?

Reply: This is a very relevant question. We now added a sentence L388-389 to explicitly say that convection and degassing mechanisms can occur at various levels in the magma plumbing system. However as we have tried to explain in the discussion section “*Evidence for shallow magma convection*”, the excess thermal radiation observed from the 15-year MODIS dataset presented in Fig. 5 suggests that the magma volume radiating thermally largely exceeds the actual erupted magma volume. This strongly suggests that magma must be convecting at shallow depth in order to convey the thermal energy efficiently. In the case of convection within a larger deeper chamber, much of the thermal energy would be lost as the gas travels several kilometers up the conduit. Indeed, results of seismic tomography exclude the presence of a large magma chamber in the shallow crust (above 6 km b.s.l.), which is agreement with most studies (see Berger et al. 2011 and references therein, which has now been added to the manuscript as ref. 70). A thermodynamic model to calculate the heat transfer and loss by the gas would have to be performed to confirm this. Although this is out of the scope of the present paper, we have now added a sentence L391-393 to mention this explicitly.

Berger, P., Got, J. L., González, C. V. & Montéiller, V. Seismic tomography at Popocatépetl volcano, Mexico. Journal of Volcanology and Geothermal Research 200, 234–244 (2011).

Is 40-70% bubbles (line 270) realistic in a magma that seems very fluid? Even with the Methods, these questions remain.

Reply: We have now added 2 additional references L299-302 to further argue that it is realistic. The first is a study of vesicularity distribution in juvenile clasts at Popocatépetl (Cross et al. 2012). The second, is a study in a more basaltic system where magma is more fluid, which shows that the vesicularity textures of ejected lapilli show ~40–76 vol% vesicles (Lautze & Houghton 2005).

Cross, J. K., Roberge, J. & Jerram, D. A. Constraining the degassing processes of Popocatépetl Volcano, Mexico: A vesicle size distribution and glass geochemistry study. Journal of Volcanology and Geothermal Research 225–226, 81–95 (2012).

Lautze, N. C. & Houghton, B. F. Physical mingling of magma and complex eruption dynamics in the shallow conduit at Stromboli volcano, Italy. Geology 33, 425–428 (2005).

In conclusion, the scientific background is present but due to its current form, I am not sure that the article can be published in Nature Communication.

Reply: We truly hope that the various modifications applied to the manuscript have improved it, and have removed the doubts the reviewer had.

Reviewers' Comments:

Reviewer #1:

Remarks to the Author:

The revised version of the manuscript "Dome cycles and crater deepening at Popocatepetl reveal convection surges of a depressurizing magma column" by Valade et al. has improved considerably. All my previous comments have been addressed satisfactorily, and some important technical and interpretative issues have been clarified.

Considering the relevance of this study to better understand the governing process of the repetitive dome cycles at Popocatepetl volcano and similar volcanoes, the high-quality data, sound methodology, and convincing interpretation, I am happy to recommend this manuscript for publication in Nature Communication in its current form.

Reviewer #2:

Remarks to the Author:

The authors have addressed my comments and other reviewers' comments thoroughly. From my point of view, the paper is ready for acceptance in Nature Communications after some minor modifications:

- Avoid using the term "conservative" to justify the use of the factor 1/3 to estimate the tephra volume. To me, it is unclear what the authors mean with "conservative" in that context. The factor 1/3 is just used as a rule-of-thumb based on a previous study, and it needs to be taken with precaution.

- Many references linking tremor with by fluid flow are still missing. Some of these references are below:

Girona, T., Caudron, C., Huber, C. (2019). Origin of shallow volcanic tremor: The dynamics of gas pockets trapped beneath thin permeable media. *J. Geophys. Res.*
<https://doi.org/10.1029/2019JB017482>

Balmforth, N. J., Craster, R. V., & Rust, A. C. (2005). Instability in flow through elastic conduits and volcanic tremor. *Journal of Fluid Mechanics*, 527, 353–377.
<https://doi.org/10.1017/S002211200400280>

Bercovici, D., Jellinek, A. M., Michaut, C., Roman, D. C., & Morse, R. (2013). Volcanic tremors and magma wagging: Gas flux interactions and forcing mechanism. *Geophysical Journal International*, 195, 1001–1022. <https://doi.org/10.1093/gji/ggt277>

Chouet, B. (1996). Long-period volcano seismicity: Its source and use in eruption forecasting. *Nature*, 380, 309–316. <https://doi.org/10.1038/380309a0>

Fujita, E., Araki, K., & Nagano, K. (2011). Volcanic tremor induced by gas-liquid two-phase flow: implications of density wave oscillation. *Journal of Geophysical Research*, 116, B09201.
<https://doi.org/10.1029/2010JB008068>

Hellweg, M. (2000). Physical models for the source of Lascar's harmonic tremor. *Journal of Volcanology and Geothermal Research*, 101, 183–198. [https://doi.org/10.1016/S0377-0273\(00\)00163-3](https://doi.org/10.1016/S0377-0273(00)00163-3)

Jellinek, A. M., & Bercovici, D. (2011). Seismic tremors and magma wagging during explosive volcanism. *Nature*, 470, 522–526. <https://doi.org/10.1038/nature09828>

Reviewer #3:

Remarks to the Author:

The article has been significantly improved. The title is more in line with the content and the content is more scientifically rigorous. I thank the authors for their very clear answers to all the questions I asked.

The results section is clear and objective, and I have no particular remarks. The data clearly show that the magma level varies over time and they show correlations with thermal fluxes and gases.

To me, however, the main problem of the paper remain the discussion, which is very speculative and proposes a model compatible with the observation but that cannot be proved. Moreover, several points are not detailed enough to be understood by a broad audience.

(1) I still do not understand how the authors determine the crater enlargement (line 345) as a function of magma rate (DRE magma flux, line 334, or degassing rate, line 345, it is not clear). Lines 331-332, it seems that the vent does not change in size. Are the authors using equation 32 from their reference 57 (Girona et al., 2014)? If so, is the viscoelastic approach appropriate for the upper and probably quite cold and brittle part of the crater? Is the diameter only determined by the model of ref 58 (Stevenson and Blake, 1998)? In this case, how do the author explain the enlargement? To what I understand, ref 58 uses a conduit diameter to calculate a magma convection but does not calculate an enlargement. The link between the "depressurization", line 312, and the crater enlargement must be explained clearly. It seems to me that a mechanical model is missing. Images of Fig 6a3 and 6a4 look similar for a very different scheme and it is confusing. I have read and re-read this part and I cannot understand what was done. I suggest either to explain the connection clearly (with sufficient detail about the assumptions used and the limitations of the approaches), or to delete everything about the enlargement if it cannot be proven.

(2) The authors must better describe the assumptions and above all the geometry of their system. For example, without reading ref 58, it was impossible to me to understand what means the magma input flux (line 334) in a system that depressurizes.

(3) I still do not understand what arguments allow to talk about "shallow magma convection". I am not convinced by the authors 'answer: "much of the thermal energy would be lost as the gas travels several kilometers up the conduit". We need a demonstration or references of works that have studied this problem. The authors explain very objectively that thermodynamic modeling is needed to determine where the gas is coming from (lines 378-380). However, without this modeling or other sufficient demonstration, I believe that there is not enough evidence to write that the observation are explained by a shallow magma convection (line 353). I think that, if this cannot be proved, this shallow convection should not be presented as a key point of the discussion.

(4) Even the link between crater deepening and volume change related to degassing is a possible scenario, but there is no evidence that this mechanism is the only explanation.

To summarize, it is difficult to understand what concerns the dynamics of the conduit without being an expert of the model (models?) used and, for a broader audience, too little information is given in the article to have a clear idea of what has been done. The conclusions of the article are interesting but they are partly based on original and robust data (SAR images, combined SO₂ and thermal measurements) and, on assumptions about the dynamics of the conduit that are not sufficiently demonstrated (shallow convection, enlarging, link deepening/densification) or already known/proposed (more outgassing magma than extruded magma, dynamics of the plumbing system of Popocatepetl). Even if the authors use honestly "speculative", "remind", "likely", "suggest", etc., it seems to me dangerous to propose the above mechanisms in such a scientific journal without robust evidences. The article has been improved but too many questions remain and I think that the short format of Nature Communication is not adapted to the publication of this work.

Response to Referees

Nature Communications manuscript NCOMMS-22-38331 – REVIEW #2

1. Response to Reviewer #1

Reviewer #1 (Remarks to the Author):

The revised version of the manuscript “Dome cycles and crater deepening at Popocatepetl reveal convection surges of a depressurizing magma column” by Valade et al. has improved considerably. All my previous comments have been addressed satisfactorily, and some important technical and interpretative issues have been clarified. Considering the relevance of this study to better understand the governing process of the repetitive dome cycles at Popocatepetl volcano and similar volcanoes, the high-quality data, sound methodology, and convincing interpretation, I am happy to recommend this manuscript for publication in Nature Communication in its current form.

Reply: We are grateful to hear that the review process has improved the manuscript and that it is judged satisfactory by the reviewer. Further concerns were raised by one of the other reviewers, so the manuscript has been further modified to try to improve it. The modifications however do not change the core message of the publication, but rather clarify/simplify aspects of the model, and remove elements of the discussion which were judged too speculative.

2. Response to Reviewer #2

Reviewer #2 (Remarks to the Author):

The authors have addressed my comments and other reviewers' comments thoroughly. From my point of view, the paper is ready for acceptance in Nature Communications after some minor modifications:

- Avoid using the term "conservative" to justify the use of the factor 1/3 to estimate the tephra volume. To me, it is unclear what the authors mean with "conservative" in that context. The factor 1/3 is just used as a rule-of-thumb based on a previous study, and it needs to be taken with precaution.

Reply: It is true that the term “conservative” could be interpreted in various ways. We have consequently replaced it by the term “approximation”.

- Many references linking tremor with by fluid flow are still missing. Some of these references are below.

Reply: The origin of tremor is indeed a prolific question, however due to the limited number of citations allowed in this journal (max. 70), we had restricted ourselves to the reference studies

reporting tremor at Popocatépetl. Nonetheless we agree that a tremor-dedicated reference was missing. We have therefore added the latest publication suggested in the list, as it describes a model which appears realistic at Popocatépetl:

Girona, T., Caudron, C., Huber, C. (2019). Origin of shallow volcanic tremor: The dynamics of gas pockets trapped beneath thin permeable media. J. Geophys. Res. <https://doi.org/10.1029/2019JB017482>

3. Response to Reviewer #3

Reviewer #3 (Remarks to the Author):

The article has been significantly improved. The title is more in line with the content and the content is more scientifically rigorous. I thank the authors for their very clear answers to all the questions I asked.

The results section is clear and objective, and I have no particular remarks. The data clearly show that the magma level varies over time and they show correlations with thermal fluxes and gases.

To me, however, the main problem of the paper remain the discussion, which is very speculative and proposes a model compatible with the observation but that cannot be proved. Moreover, several points are not detailed enough to be understood by a broad audience.

We truly thank the reviewer for taking the time to review the paper once again, and to provide this harsh, but honest and justified criticism. We have taken this very seriously. We have addressed the concerns raised in a conscientious way, and now truly believe that the newly revised manuscript has much improved in quality and clarity thanks to these.

The main corrections applied to the manuscript are summarized in the reply to the reviewer's last comment. All specific comments have been answered in detail below, in which we underline how the manuscript has been modified. (Please note that the line numbers Lxxx used throughout our answers refer to the revised "track-change" manuscript, in order to easily visualize how the manuscript was edited).

(1) I still do not understand how the authors determine the crater enlargement (line 345) as a function of magma rate (DRE magma flux, line 334, or degassing rate, line 345, it is not clear). Lines 331-332, it seems that the vent does not change in size. Are the authors using equation 32 from their reference 57 (Girona et al., 2014)? If so, is the viscoelastic approach appropriate for the upper and probably quite cold and brittle part of the crater? Is the diameter only determined by the model of ref 58 (Stevenson and Blake, 1998)? In this case, how do the author explain the enlargement? To what I understand, ref 58 uses a conduit diameter to calculate a magma convection but does not calculate an enlargement. The link between the "depressurization", line 312, and the crater enlargement must be explained clearly. It seems to me that a mechanical model is missing. Images of Fig 6a3 and 6a4 look similar for a very

different scheme and it is confusing. I have read and re-read this part and I cannot understand what was done. I suggest either to explain the connection clearly (with sufficient detail about the assumptions used and the limitations of the approaches), or to delete everything about the enlargement if it cannot be proven.

Reply: This first comment with multiple sub-questions calls for clarification regarding the “enlargement”. It mainly derives from the confusion between the “inner crater diameter” (which indeed enlarges, see satellite observations Fig. 2b), and the “conduit diameter” (which is assumed constant, and which is essentially recovered from Stevenson and Blake 1998). We give below a detailed answer to each sub-question, and explain how we clarified each point in the manuscript:

- *"I still do not understand how the authors determine the crater enlargement (line 345) as a function of magma rate (DRE magma flux, line 334, or degassing rate, line 345, it is not clear)."*

Reply: The crater enlargement is *measured* from satellite data (see Fig. 2b), it is *not* calculated from the model. Only the magma level in the conduit is recovered from the model, which we assume to be a proxy for the *crater depth*. As such, we understand that the sentence (former line 345) led to confusion, because we only model the crater deepening, not the crater enlargement. Instead, this enlargement is thought to be a consequence of the deepening, due in particular to the successive landslides suffered by the inner crater walls which are left unstable (an example is shown in Supp. Fig. 6). We have corrected the sentence by removing the term “enlargement” when commenting on the model's conclusions (L401), and we have added a new sentence to clearly state that this *observed* crater enlargement could be a consequence of the *modeled* column deepening (L401-404).

Regarding the confusion about what the model uses as input data (i.e., “DRE magma flux”, or “degassing rate”): the confusion arises from the fact that the two are linked (as formerly stated line 564), and so we used both terms interchangeably. Indeed, the model used here (Girona et al. 2014) uses the scenario in which degassing is coupled with magma convection in the conduit, as parameterized by Stevenson and Blake 1998. The latter state that “*gas flux depends on the volumetric magma upflow rate (Q) and the amount of gas lost*”, expressed in m^3/s . Our approach is therefore the following: from the measured SO_2 flux [kg/s] we compute a DRE magma input rate [m^3/s] (i.e., Q_{degas} ranging from 1-10 m^3/s in Fig. 5, see Methods section “Magma budget estimation”), which we then feed to the model of Girona et al. 2014 together with a value for the amount of gas lost (n_c) taken from a former study at Popocatépetl by Witter et al. 2005. We have now given a more explicit explanation of this in both the Discussion L370-375 and Methods L674-680.

- *"Are the authors using equation 32 from their reference 57 (Girona et al., 2014)? If so, is the viscoelastic approach appropriate for the upper and probably quite cold and brittle part of the crater? Is the diameter only determined by the model of ref 58 (Stevenson and Blake, 1998)? In this case, how do the author explain the enlargement? To what I understand, ref 58 uses a conduit diameter to calculate a magma convection but does not calculate an enlargement."*

Reply: This question was partly answered in the previous question: we are *not* modeling

crater enlargement. We have now restructured the discussion paragraph dedicated to the long-term crater deepening, to be clear about 1) what the model geometry/assumptions/inputs are (i.e. better explain the “conduit diameter”, the degassing rates, etc.), 2) what we are modeling and which conclusions can be drawn (i.e., clear distinction between the crater deepening and crater widening). The methods section has also been further detailed to provide a thorough description of the key aspects of the model, and the Fig.7 has been modified to reflect these clarifications.

Detailed explanation to the reviewer: The model parametrization indirectly defines the *conduit diameter* (not the *crater diameter*). It is correct that Stevenson and Blake 1998 (former ref 58) relates the conduit diameter to the magma convection (to be exact, it relates the conduit radius to the magma upflow rate), and that Girona et al. 2014 (former ref 57) directly implements this equation in the model scenario contemplating convection in the conduit. We do *not* use equation 32 of ref. 57, but the compact equation for the depressurization rate ΔP (equ. 40), which encompasses and simplifies several differential equations (including equ. 32). This pressure variation ΔP is calculated for 3 distinct DRE magma input fluxes ($Q = 1, 7,$ and $10 \text{ m}^3/\text{s}$, held constant through time), which according to ref's 58 convection parameterization, equate to conduit radii $R = 8.3, 13.5,$ and 14.8 m , respectively. This is now explicitly stated in the Discussion L374-377 and Methods L680-683.

- *"The link between the "depressurization", line 312, and the crater enlargement must be explained clearly. It seems to me that a mechanical model is missing."*

Reply: We have now clarified in the discussion (L400-404) that only the crater deepening is related to the depressurization, and that the crater enlargement is only a consequence of this deepening. Only the crater deepening is compared to the model output (Fig. 7d), so we believe that introducing a mechanical model would be outside the scope of this paper.

- *"Images of Fig 6a3 and 6a4 look similar for a very different scheme and it is confusing. I have read and re-read this part and I cannot understand what was done. I suggest either to explain the connection clearly (with sufficient detail about the assumptions used and the limitations of the approaches), or to delete everything about the enlargement if it cannot be proven."*

Reply: We have modified the sketch in Fig 6a4 (now Fig 6d) to better represent the ongoing processes, and to make a smoother transition from Fig.6a3 (now Fig 6c). We have also changed the TSX image below Fig 6a4 (now Fig 6d): the image acquired 2016-12-29 has been replaced by the acquisition on 2017-02-11 (i.e. ~45 days afterwards), in which the piston-collapse structures are more obvious. Moreover, sketches Fig. 6 a-c have been simplified in order to remove aspects which were too speculative (convection arrows, widening of the conduit diameter, etc.).

As previously commented, the crater enlargement is a satellite observation (see Fig. 1b), not a calculation from the model. However it is true that this observation is not obvious on short time-scales but rather on long time-scales, so we have removed this from Fig 6 (i.e., now focused on the short-term dome evolution), and only kept it in Fig 7 (i.e., now focused on the

long-term crater evolution).

(2) The authors must better describe the assumptions and above all the geometry of their system. For example, without reading ref 58, it was impossible to me to understand what means the magma input flux (line 334) in a system that depressurizes.

Reply: We have now reformulated the discussion section dedicated to the model, so that the reader would have a clear statement of the key assumptions and geometry of the model. We have also complemented the Methods section, in order to give thorough details on the model parameterization (these details were judged unnecessary and cumbersome in the discussion, and better suited in the Methods). The sketch of the system geometry, along with the values of the corresponding parameters, are kept in Supp. Table 1 (minor improvements were made in order for it to be more exhaustive).

(3) I still do not understand what arguments allow to talk about "shallow magma convection". I am not convinced by the authors 'answer: "much of the thermal energy would be lost as the gas travels several kilometers up the conduit". We need a demonstration or references of works that have studied this problem. The authors explain very objectively that thermodynamic modeling is needed to determine where the gas is coming from (lines 378-380). However, without this modeling or other sufficient demonstration, I believe that there is not enough evidence to write that the observation are explained by a shallow magma convection (line 353). I think that, if this cannot be proved, this shallow convection should not be presented as a key point of the discussion.

Reply: Although we are convinced that only shallow convection can explain the range of observations presented (i.e., the extraordinarily high excess degassing and excess thermal radiation, the very active surface activity (in 8 years of TSX acquisitions, *never* has the crater floor appeared the same between two images)), we agree that at the moment, we do not have the thermodynamic model necessary to robustly and unequivocally prove how shallow the convection is. Demonstrating this would in our opinion require an entire study dedicated to this aspect, with the appropriate thermodynamic and degassing models, both involving further assumptions and adequate data analysis. We have judged that incorporating this into the present paper would have diluted the take-away message, which is mainly focused on the short-term dome construction-destruction cycles. As a consequence, we have removed all mentions of "shallow convection" from the paper, which were at this stage too speculative:

- *title*: changed to remove the term "convection" and focus on the lava dome cycles
- *abstract*: edited
- *discussion*: removed the entire paragraph dedicated to the shallow convection

(4) Even the link between crater deepening and volume change related to degassing is a possible scenario, but there is no evidence that this mechanism is the only explanation.

Reply: We are a bit puzzled by this comment. Indeed, the paragraph starts by acknowledging (L339) that: *"Two processes can be considered: i) the excavation due to repetitive explosions, and/or ii) the depressurization of the shallow plumbing system due to persistent passive*

degassing and decreasing magma supply rates". It continues by stating that *"Although both mechanisms likely operate and contribute to the progressive crater deepening, our observations suggest that the latter plays an important role which was previously unsuspected"*. This claim is then supported by 2 supplementary figures (Supp. Fig. 8 and 9) which show that although the first mechanism is likely the most obvious, it does not have clear data supporting that it is the main driver. The model which is then applied, is merely intended to show that within reasonable assumptions (taken from the literature and/or our data), our observations can be explained by such a mechanism. Of course we do not rule out other mechanisms, but we find that it is a relevant finding to show that an unsuspected mechanism could contribute to this new observation describing the decadal deepening of the crater. Nonetheless, we understand the critic, and we do not want this model to take all the attention from the discussion as it is only a part of the story. We have therefore re-arranged the Discussion and the Figures 6 and 7, in order to better separate the short-term dome cycles, and the long-term crater deepening. This has in our opinion simplified and clarified the take-away message.

To summarize, it is difficult to understand what concerns the dynamics of the conduit without being an expert of the model (models?) used and, for a broader audience, too little information is given in the article to have a clear idea of what has been done. The conclusions of the article are interesting but they are partly based on original and robust data (SAR images, combined SO₂ and thermal measurements) and, on assumptions about the dynamics of the conduit that are not sufficiently demonstrated (shallow convection, enlarging, link deepening/densification) or already known/proposed (more outgassing magma than extruded magma, dynamics of the plumbing system of Popocatepetl). Even if the authors use honestly "speculative", "remind", "likely", "suggest", etc., it seems to me dangerous to propose the above mechanisms in such a scientific journal without robust evidences. The article has been improved but too many questions remain and I think that the short format of Nature Communication is not adapted to the publication of this work.

Reply: All reviewers have acknowledged that the presented observations are unique and robust, and therefore we want to believe that they are worthy of the broad readership offered by this journal, provided that the righteously formulated criticisms are adequately addressed. Indeed, this final summary comment clearly shows that our discussion had failed to: 1) properly distinguish the discussion elements with which we are confident, and those which are more speculative / model-based, and 2) make the model understandable for a broad audience (requiring simplification/clarification in the discussion), while being sound and complete for an expert (requiring more details in the methods). Rather than complexifying the paper with additional models (i.e., thermodynamic model, mechanical model), which we thought would have "diluted" the take-away message of the paper, we have decided to restructure the discussion, and simplify/clarify where needed. Here is a summary of the main changes applied to the manuscript:

- (1) We have provided a clearer and more detailed explanation of the long-term deepening model in both the Discussion and Methods sections, with particular attention to providing

a clear description of what are *data*, *assumptions*, and *outputs*. We have taken special care to clarify that the enlargement refers to the “crater” and not the “conduit” (the former being a satellite observation, while the latter is held constant in the model and constrained from the data), and that only the crater deepening is modeled and compared to the observations. The description of the key assumptions and chosen values are now better structured and more thorough, and should allow anyone to have a clear understanding of what is being done. Importantly, it should now be clear that the model is merely intended to show that within reasonable assumptions (exclusively derived from either previous work at Popocatepetl and/or from the presented data), the crater deepening observations can be explained by a mechanism which is usually not considered, which we find noteworthy (with all due precautions).

(2) We have removed speculative aspects from the paper, and in particular the notion of “shallow convection”: the paragraph “Evidence for shallow magma convection” was entirely removed from the Discussion, and the Title, Abstract and Figures have been edited accordingly. It now makes the manuscript more focused on the unique and robust observations presented here, which concern both the short-term dome construction-subsidence cycles, and the long-term crater deepening.

(3) Figures 6 and 7 have been rearranged and respectively improved, in order to separate the short- and long-term observations / model / interpretations, and thereby simplify the take-away message.

- Fig. 6: we simplified the sketches to remove the speculative aspects (i.e., convection arrows, increasing conduit width at depth), modified the last sketch/image to better understand the transition, added a subplot to show that the short-term model can explain the observed dome subsidence trends (i.e., this subplot is an improvement of the former Fig.7a).

- Fig. 7: we have clearly distinguished the driving processes thought to explain the crater deepening and widening, we have edited the sketches to reflect the geometry of the model (conduit connected to a reservoir, both submitted to the same pressure changes), we have removed the subplot depicting the model ΔP calculation (former Fig.7b) in order to simplify the take-away message and be more coherent with Fig.6e (the values of ΔP are kept as secondary y-axis in Fig.7d), and we have added a secondary time-axis to easily refer to observed crater depths.

These modifications (which we have tried to keep to a minimum since 2 reviewers had accepted the paper), have in our opinion truly improved the manuscript, so we are very grateful to the reviewer for this. Importantly, we hope that the reviewer’s concern regarding the lack of clarity of the model are now dispelled, and that this well-founded criticism has been properly addressed. In doing so, we hope that the model will no longer overshadow the rest of the paper, as it is only “a part of the story” described in the manuscript.

We strongly hope that the reviewer will be satisfied with this revised version of the manuscript. With renewed sincere gratitude and on behalf of all co-authors, we send our best regards.

Reviewers' Comments:

Reviewer #3:

Remarks to the Author:

The revised version of the manuscript "Lava dome cycles reveal rise and fall of magma column at Popocatepetl volcano" by Valade et al. has been improved and is clearer, particularly the method section. If the editor thinks that the article, in its current form, is detailed enough and is suitable for a wide audience, I agree with the other reviewers and recommend the article for publication in Nature Communication.

Response to Referees

Nature Communications manuscript NCOMMS-22-38331 – REVIEW #3

Reviewer #3 (Remarks to the Author):

The revised version of the manuscript “Lava dome cycles reveal rise and fall of magma column at Popocatépetl volcano” by Valade et al. has been improved and is clearer, particularly the method section. If the editor thinks that the article, in its current form, is detailed enough and is suitable for a wide audience, I agree with the other reviewers and recommend the article for publication in Nature Communication.

Reply: We are grateful to hear that the review process has improved the manuscript and that it is judged satisfactory by the reviewer.